# Novel Black Seed Polysaccharide Extract-g-Poly (Acrylate) pH-Responsive Hydrogel Nanocomposites for Safe Oral Insulin Delivery: Development, In Vitro, In Vivo and Toxicological Evaluation

**DOI:** 10.3390/pharmaceutics15010062

**Published:** 2022-12-25

**Authors:** Farya Shabir, Asif Mahmood, Nadiah Zafar, Muhammad Zaman, Rai Muhammad Sarfraz, Hira Ijaz

**Affiliations:** 1Faculty of Pharmacy, The University of Lahore, Lahore 54000, Pakistan; 2Department of Pharmacy, University of Chakwal, Chakwal 48800, Pakistan; 3Faculty of Pharmacy, University of Central Punjab, Lahore 54000, Pakistan; 4College of Pharmacy, University of Sargodha, Sargodha 40100, Pakistan; 5Department of Pharmaceutical Sciences, Pak-Austria Fachhochschule Institute of Applied Sciences and Technology, Mang, Khanpur Road, Haripur 22620, Pakistan

**Keywords:** pH-responsive hydrogel nanocomposites, black seed polysaccharide extract, montmorillonite-sodium, insulin oral delivery, Ins-BA hydrogel, Ins-Mmt-BA hydrogel NC

## Abstract

Oral delivery of insulin has always been a challenging task due to harsh gut environment involving variable pH and peptidase actions. Currently, no Food and Drug Administration (FDA) approved oral insulin formulation is commercially available, only intravenous (IV) or subcutaneous (SC) routes. Therefore, it is really cumbersome for diabetic patients to go through invasive approaches for insulin delivery on daily basis. In the present study, a novel pH-responsive hydrogel nanocomposite (NC) system was developed and optimized for safe oral delivery of insulin. Black seed polysaccharide extract-based hydrogel (BA hydrogel) was formulated by free radical polymerization and loaded with insulin. Blank BA hydrogel was also incorporated with insulin-loaded montmorillonite nanoclay (Ins-Mmt) to form an Ins-Mmt-BA hydrogel NC and compared with the insulin-loaded hydrogel. Swelling, sol-gel analysis and in vitro release studies proved that Ins-Mmt-BA6 hydrogel NC has the best formulation, with 96.17% maximum insulin released in 24 h. Kinetic modeling applied on insulin release data showed the Korsemeyer-Peppas model (R^2^ = 0.9637) as the best fit model with a super case II transport mechanism for insulin transport (n > 0.89). Energy Dispersive X-ray (EDX) Spectroscopy, Fourier Transformed Infrared (FTIR) spectroscopy and Powdered X-ray diffraction (PXRD) analysis results also confirmed successful development of a hydrogel NC with no significant denaturation of insulin. Toxicity results confirmed the safety profile and biocompatibility of the developed NC. In vivo studies showed a maximum decrease in blood glucose levels of 52.61% and percentage relative bioavailability (% RBA) of 26.3% for an Ins-Mmt-BA hydrogel NC as compared to BA hydrogels and insulin administered through the SC route.

## 1. Introduction

Continual research in the field of life sciences is going on to improve quality of life. In order to achieve this task, persistent efforts are being made in terms of development of new therapeutic drugs and drug delivery systems. An improved pharmacokinetic profile of a targeted drug is the key for increasing patient compliance and to gain trust of patients on that particular dosage form. Patients’ compliance is also desired in ailments requiring prolonged therapies, i.e., diabetes, hypertension, carcinomas, etc.

Diabetes mellitus (DM) is an endocrine disorder either due to impaired insulin release, insufficient action or both, resulting in persistent hyperglycaemia. According to estimates of the International Diabetes Federation (IDF), the world diabetic population in 2019 was 463 million and could be up to 700 million by 2045 [1]. Such a massive number of patients worldwide often require long-term insulin therapy via invasive parenteral routes, especially patients with DM Type 1 (DM1). DM1 is a chronic autoimmune disease leading to selective destruction of pancreatic β-cells involved in the production of insulin within the body [2]. The clinical onset of the disease reflects end stage destruction of β-cells, leading to DM1 [3]. For DM1 patients, the currently available dosage forms are either subcutaneous/intravenous (SC/IV) insulin injections or SC insulin pumps and pens. Therefore, it is a dire need to find and introduce some non-invasive routes such as oral routes and novel drug delivery approaches for safe prolonged delivery of insulin [4].

Humulin^®^ R U-100 (containing human insulin with the strength 100 units/mL) is a sterile, colourless and aqueous solution formulation. It is a short-acting polypeptide hormone produced by an *Escherichia coli* strain obtained by using recombinant DNA technology. Firstly, it controls blood sugar by inhibiting hepatic glucose production, and secondly, it stimulates glucose uptake from peripheral skeletal muscles and fat cells. When administered subcutaneously (SC), it starts showing therapeutic response within 30 min and its effect lasts up to 8h, with the dose in the range of 0.05–0.4 units/kg. With intravenous (IV) administration, the pharmacologic effect of the drug starts within 10–15 min and lasts up to 4h with the dose range of 0.1–0.2 units/kg. However, the IV route of administration requires strict clinical monitoring [5].

The major hurdle for insulin delivery via oral route is the extremely erratic pH of the gut and exposure to gastric peptidases that can destroy many proteinaceous drugs such as insulin [6]. By using some pH-responsive drug delivery systems that restrict the insulin release in the stomach and allow it to release its contents at intestinal pH, this loss can be controlled. Hence, the chances of insulin absorption via oral route can be pronounced [7]. Earlier, efforts have been made to deliver insulin via oral route by developing delivery systems such as hydrogel microparticles, solid-lipid nanoparticles, nanocomposites, pH-responsive hydrogels, etc., [8,9,10,11,12]. However, these drug delivery systems still are far away from clinical trials. Currently, no Food and Drug Administration (FDA) approved oral insulin formulation is commercially available [13]. Therefore, this is a topic of great interest to develop a novel drug delivery system for safe oral delivery of insulin [7,14].

Currently, nanomaterials are widely employed in the pharmaceutical sector due to their small-size-related benefits. Nanomaterials possess variable physical, mechanical, optical, magnetic, etc., properties due to their nano size as compared to their bulk counterparts. These include high surface area, development of magnetism, increased thermal and electrical conductivity, improved mechanical strength and showing anti-microbial activity [15]. A few examples of nanomaterials are graphite, carbon nanotubes (CNTs), nanocellulose, magnetic nanoparticles (NPs), metallic NPs and clay NPs (montmorillonite-Na^+^ clay, bentonite, attapulgite, kaolin, etc.). Montmorillonite-Na^+^ (Mmt-Na^+^) is a largely used nanoclay with a layered structure having the capability to incorporate many drugs and proteins inside their layered geometry [16].

Hydrogels are interesting drug delivery systems, as their three-dimensional polymeric structure can imbibe large amounts of water through swelling under specific conditions such as stimuli-responsive hydrogels. Hydrogel nanocomposites (NCs) are emerging drug delivery systems involving a combination of hydrogel and nanomaterials through different techniques. For example, clay NC hydrogels, metallic NC hydrogels, carbon nanotubes and graphene-based NC gels, hydroxyapatite NC gels and magnetic NC hydrogels, etc. Different methods to prepare NC hydrogels include crosslinking of hydrogel in the presence of nanomaterials, in situ incorporation of drug-loaded nanomaterials within crosslinked hydrogels, incorporation of NPs within crosslinked hydrogels via inward diffusion and co-dependent NC hydrogels having NPs as crosslinkers. Traditional hydrogels have low mechanical strength and limited functionality. Meanwhile, NC hydrogels offer dual benefits of hydrogels such as biocompatibility, targeted release, low toxicity, etc., and of nanomaterials such as improved physical and mechanical properties and imparting of unique properties (electric conductivity, magnetic responsiveness, anti-bacterial and anti-oxidant potentials, etc.) [17,18]. Hence, there are many polymeric combinations with poor mechanical properties that can be modulated as NC hydrogels.

In previous studies, the use of several natural and synthetic biodegradable polymeric systems has been attempted to enhance oral bioavailability of insulin. Natural polymers such as chitosan, alginate and their derivatives, poly-g-glutamic acid (g-PGA) and starch-based nanoparticles, were prepared for oral insulin delivery. Many synthetic polymers such as poly(lactide-co-glycolide) (PLGA), pluronic/poly(lactic acid) (PLA) vesicles and poly-caprolactone (PCL)-based systems for oral insulin delivery have been tried. However, none of them have reached the level of clinical trials. Much work still has to be done in this regard [9].

In this study, we tried to develop a pH-responsive hydrogel NC for safe oral delivery of insulin. In the first step, pH-responsive acrylic acid (AA)-grafted black seed hydrogels (BA) were developed. The polymer used to formulate the hydrogel was a water-soluble natural polysaccharide obtained from *Nigella sativa* seeds (commonly known as black cumin or black seed). Insulin intercalated montmorillonite sodium clay (Ins-Mmt) was loaded in previously developed hydrogels by an inward diffusion technique to make pH-responsive hydrogel NCs (Ins-Mmt-BA). BA hydrogel formulations were also directly loaded with insulin for comparison with Ins-Mmt-BA hydrogel NCs. Neither the pH-responsive hydrogel with this polymer extract nor such pH-responsive hydrogel NC were made before or tried for insulin oral delivery. Developed direct insulin-loaded hydrogels and insulin-loaded hydrogel NCs were compared for in vitro characterizations, in vivo evaluation and acute oral toxicity studies.

## 2. Experimental

### 2.1. Materials

*Nigella sativa* seeds (black seed/kalonji) were obtained from a regional market of Lahore, Pakistan. Montmorillonite sodium (Mmt-Na^+^) nanoclay (with a cation-exchange capacity of 92 mEq/100g) and acrylic acid were procured from Sigma-Aldrich, Germany. *N, N*-methylene bisacrylamide (MBA) was obtained from Fluka of Switzerland. Hydrochloric acid (HCl), *n*-hexane and absolute ethanol were obtained from Riedel-de Haen of Germany. Sodium dihydrogen phosphate (NaH_2_PO_4_), disodium hydrogen phosphate (Na_2_HPO_4_) and potassium persulphate (KPS) were purchased from Sigma-Aldrich, Hamburg, Germany. Injection *Humulin*-*R^®^* (1 Vial × 10 mL with strength 100 IU/mL) of Eli-Lilly *Pakistan* (Pvt) Ltd. (Karachi, Pakistan) was purchased from a local pharmacy of Lahore, Pakistan. Sodium sulphate (Na_2_SO_4_), HPLC grade acetonitrile (ACN), HPLC grade methanol and alloxan monohydrate (C_4_H_2_N_2_O_4_.H_2_O) were also purchased from Sigma-Aldrich, Germany. The University of Lahore (UOL) provided the albino rabbits as experimental animals. All the chemicals utilized in the experiment were of analytical grade. Freshly prepared distilled water (Ph.D. Research Lab of The University of Lahore, Lahore, Pakistan) was utilized in this experiment.

### 2.2. Extraction of Black Seed Polysaccharide Extract (BSPE)

The method used for extraction was the same as by Trigui et al. (2018) with little modification [19]. A total of 100 g of *Nigella sativa* seeds were ground coarsely after manual cleaning and screening. The grinded seeds were pre-extracted with sufficient quantity of ethanol 95% solution at room temperature to remove any small molecules and impurities. The residue hence obtained was treated twice at 90 °C with deionized water (30 times the volume of the residue). A hot air oven operated at 40 °C was employed to dry the filtrate. The dried extract was stored in air-tight containers for use in future studies. The percentage yield of the BSPE was calculated by the following formula.
$$Y\;\left(\%\right)=\frac{W_{1}}{W_{o}}\times 100$$

In this formula, *W_1_* was the weight of the extracted BSPE and *W_o_* was the weight of the dried sample. The water-soluble extract of black seed has a carbohydrate content of 71.50% with carbohydrate distribution of galacturonic acid (30.20%), glucuronic acid (17.66%) and neutral sugars mainly consisting of arabinose (5.83%) and galactose (5.76%) [19].

### 2.3. Process of Ins-Mmt-BA Hydrogel Nanocomposite Development

The Ins-Mmt-BA hydrogel NCs were developed in the following three main steps.

#### 2.3.1. Development of BA Hydrogels

In the first step, different formulations of black seed polysaccharide extract-g-poly (acrylate) hydrogels (BA1-BA6) were developed (Table 1). For this purpose, a weighed quantity of BSPE was dissolved in a sufficient quantity of distilled water (1% *w*/*v*) under continual stirring at room temperature. The initiator KPS was added in weighed quantity to the BSPE solution to generate free radicals and active sites at the polymeric backbone. A second solution was prepared by dissolving a calculated amount of crosslinker MBA in a measured volume of AA monomer. This second solution was then transferred to the polymer/initiator solution dropwise. The final volume was adjusted by adding a measured amount of distilled water. This reaction mixture was then shifted into test tubes, which were tightly covered with aluminium foil. These test tubes were then sonicated for 5 min to remove any dissolved gas bubbles. After that, the test tubes were transferred to a digital water bath, which was maintained at 60 °C over a period of 24 h. After 24 h, the transparent hydrogel rods were obtained by carefully breaking the glass tubes. The hydrogel rods were then cut into 5 mm discs with the help of a sharp knife. The hydrogel discs were washed with an ethanol:water (30:70) mixture to remove any excess or unreacted reagents. After washing, the hydrogel discs were removed and put into large petri dishes for drying in the hot air oven at 50 °C. The dried hydrogel discs were stored in well-closed containers for future study and developing hydrogel NCs.

#### 2.3.2. Preparation of Insulin Intercalated Mmt-Na^+^ (Ins-Mmt) Nanoclay

The second step in the development of Ins-Mmt-BA hydrogel NCs was loading or intercalation of insulin in between montmorillonite sodium (Mmt-Na^+^) nanoclay layers. The Mmt-Na^+^ clay possesses a unique layered structure with tuneable inter-layer distance. It also has hydrophilic and hydrophobic regions, thus allowing incorporation of a number of different drugs and proteins in between these layers [20,21]. A total of 1 g of Mmt-Na^+^ was suspended in 500 mL of distilled water for 2 h with continuous stirring. The pH of this suspension was adjusted to 2 by adding dilute acidic solution dropwise. This is because insulin is difficult to dissolve in water. Furthermore, the Mmt-Na^+^ is a cation exchanger and insulin carries a positive charge at pH below 5.4. During the entire process, the pH of the system was maintained at 2 due to its significant impact on intercalation. The stability of insulin remains unaffected even at low pH because in earlier literature it has been proven that the native fold of insulin remains intact at pH as low as 2, and it is still capable of forming dimers to those present in hexameric insulin structure at high pH [22].

About 100 mg of insulin (10% by weight of Mmt-Na^+^) was added in the above-mentioned Mmt-Na^+^ suspension and kept in refrigerator with frequent stirring for 3 days. After that, the product was filtered and dried at 25 °C under vacuum. Thus, insulin intercalated Mmt-Na^+^ (Ins-Mmt) was obtained. It was stored in air-tight containers and kept in a refrigerator until further use.

#### 2.3.3. Loading of Ins-Mmt into BA Hydrogels to Develop Ins-Mmt-BA Hydrogel NCs

In the last step, the Ins-Mmt nanoclay was post-loaded into the previously developed BA hydrogels to develop novel Ins-Mmt-BA hydrogel NCs. For this purpose, weighed dried BA hydrogels were kept in a measured volume of pH 7.4 phosphate buffer solution (PBS). When hydrogels were swollen near to their optimum extent at the 21st h (so that the hydrogel may offer wider pores and higher inward diffusion rate of PBS, ultimately leading to instant loading of Ins-Mmt), a weighed quantity of Ins-Mmt clay containing 60 IU of insulin was added to the PBS having swollen hydrogel discs. The mixture was gently stirred for 15 min to ensure optimum loading of Ins-Mmt into the hydrogels to form hydrogel NCs. After 15 min, the NC discs were removed, dried at 25 °C and kept in air-tight containers in a refrigerator for further studies and characterizations [23].

### 2.4. Directly Insulin-Loaded BA Hydrogel (Ins-BA) Development for Comparison

All formulations of BA hydrogels (BA1-BA6) were also directly loaded with insulin (Ins-BA1 to Ins-BA6) for comparison with their respective hydrogel NC batches. Each formulation disc was soaked in a sufficient quantity of PBS of pH 7.4 containing 60 IU of insulin. After maximum swelling, hydrogels were removed, dried at 25 °C and stored in well-closed containers in a refrigerator. Thus, insulin-loaded BA hydrogels (Ins-BA) were developed [24].

### 2.5. Performance of Swelling Studies

Swelling studies such as swelling ratio (*q*) were performed at pH 1.2 and 7.4 to find and compare pH responsiveness of the developed hydrogels and their respective hydrogel NC formulations. For these studies, both dried hydrogels and hydrogel NCs were weighed and soaked in pH 1.2 and 7.4 buffer solutions. Their swollen mass was taken out at certain intervals (0, 0.5, 1, 1.5, 2, 3, 4, 8, 10, 12, 14, 18, 24, 36, 48, 60 and 72 h). After each interval, hydrogels and NC hydrogels were drawn from buffer solutions. The excessive solution was removed from the surface and discs were weighed. These measurements were taken regularly until the achievement of swelling equilibrium (i.e., no increase in mass was observed at consecutive intervals). The formula used to measure the swelling ratio (*q*) was
$$q=\frac{Ms}{Mo}$$
where, *M_s_* is the mass of the swollen gel and NC gel formulations. Moreover, *M_o_* is the initial mass of the dried hydrogel and hydrogel NC discs [25].

### 2.6. Measurement of Drug Loading Efficiency (% DLE)

To find out the drug loading efficiency (% DLE), known weights of all BA hydrogels and Ins-Mmt-BA hydrogel NCs were ground. The ground samples were then soaked in 0.1 M PBS of pH 7.4 for 24 h. After that, soaked samples were sonicated for 15 min for complete extraction of insulin. Hydrogel and NC samples were then centrifuged (Hitachi Zentrifugen EBA 20, Hitachi Ltd., Tokyo, Japan) at 300 rpm for 10 min for removal of polymeric debris. Polymeric debris was also washed with a sufficient amount of fresh solvent to remove any adhered drug. The supernatant solution was carefully removed by micropipette for estimation of insulin by a UV-visible spectrophotometer (Pharmaspec-1700, Shimadzu, Kyoto, Japan) at ʎ_max_ = 214 nm. % DLE was determined by the following formula.
$$DLE\;\left(\%\right)=\frac{Amount\;of\;actual\;drug\;in\;dosage\;form}{Amount\;of\;drug\;added\;in\;dosage\;form}\times 100$$

Here, actual drug is the total amount of drug calculated by the UV-visible spectrometer in the dosage unit. Meanwhile, added drug is the amount of total drug that was tried to be loaded in the dosage unit by adding in swelling medium [26].

### 2.7. Sol-Gel Fraction Determination

Sol-gel fraction was determined for all BA hydrogels (BA1-BA6) and Ins-Mmt-BA hydrogel NC (Ins-Mmt-BA1 to Ins-Mmt-BA6) formulations to find out the crosslinked polymer percentage and effect of loaded Ins-Mmt on gel fraction of NCs, respectively. For sol-gel fraction determination, both the hydrogel and hydrogel NC formulation discs were individually ground to 2 mm size pieces with the help of a pestle and mortar in known quantities. The ground discs were then extracted for 4 h in a Soxhlet apparatus at 95 ± 5 °C with boiling deionized water. Soluble reactants were collected in a round bottom flask in boiling water after condensation. The hydrogel and NC networks were then removed and dried in a hot air oven at 40 °C–45 °C. Sol-gel fraction measurements were performed using the following equations.
Gel fraction (%)=100−Sol fraction
$$\mathrm{Sol}\;\mathrm{fraction}\;\left(\%\right)=\frac{W_{o}-W_{t}}{W_{o}}\times 100$$where, *W_t_* is the mass of hydrogel and hydrogel NC samples after extraction and drying. Meanwhile, *W_o_* is the weight of their dried gel and NC discs before extraction [24].

### 2.8. Fourier Transformed Infrared (FTIR) Spectroscopy

To confirm compatibility of the formulation ingredients of hydrogels and NCs, successful grafting of AA and loading of insulin FTIR-spectroscopic analysis of BSPE, AA, BA hydrogel, Mmt-Na^+^, Ins-Mmt, Ins-BA hydrogel and Ins-Mmt-BA hydrogel NC was carried out. All spectra were recorded with the IR Prestige-21 instrument (Shimadzu, Japan). For analysis of sample potassium bromide (KBr), tablets were prepared using 150 kg/cm^2^ hydraulic pressure. The scanning of all samples was performed over a scanning range of 4000–500 cm^−1^ at room temperature. The analytical data system software was LabSolutions IR [27].

### 2.9. Scanning Electron Microscopy (SEM)

To determine and compare surface morphology of BA hydrogels and Ins-Mmt-BA hydrogel NCs, SEM analysis was performed. For sample preparation, lyophilized hydrogel and NC hydrogel discs were sputtered with 10 nm gold and subsequently placed on the aluminum stub. The prepared samples were examined under an electron microscope (JSM-6490A, Tokyo, Japan) at different resolutions [28].

### 2.10. Energy Dispersive X-ray Spectroscopic (EDX) Analysis

EDX analysis was done for confirmation of loading of Ins-Mmt into hydrogel NCs. The Energy Dispersive Spectrometer (Escalab 250 Xi, Thermo Fisher Scientific, Waltham, MA, USA) was used to record the difference in elemental composition of BA-hydrogel before and after Ins-Mmt loading [29].

### 2.11. Powdered X-ray Diffraction Analysis (PXRD)

To evaluate changes in the physical form of insulin in Ins-Mmt clay, Ins-BA hydrogel and Ins-Mmt-BA hydrogel NC PXRD analysis was performed. Analysis was done by using an X-ray diffractometer of Panalytical at a scan rate of 1° min^−1^ between 3–60° angles at 2*θ* [30].

### 2.12. Differential Scanning Calorimetry and Thermogravimetric Analysis (DSC and TGA)

To assess thermal changes such as enthalpy changes, phase transition temperature and thermal stability of insulin in gel verses gel NCs, DSC and TGA were executed by Discovery SDT650, USA. Samples were placed in aluminium (Al) pans for gradual heating at a rate of 10 °C/min. Temperature was varied from ambient temperature to 500 °C. The results were obtained using TA Instruments (New Castle, DE, USA) Trios (V4.5.1.42498) software [31].

### 2.13. In Vitro Insulin Release Measurements

The measurements of insulin release from all BA hydrogels and respective hydrogel NCs were performed in vitro via the USP dissolution apparatus II (Pharma Test, Hainburg, Germany). The absorbance measurements of released insulin from test formulations were taken at λ_max_ 214 nm by a UV-visible spectrophotometer (Pharmaspec-1700, Shimadzu, Japan). Each hydrogel and hydrogel NC formulation disc was firstly placed at the bottom of the basket of the dissolution apparatus in 500 mL of 0.1 N HCl of pH 2. After 2 h, the same discs were transferred to pH 7.4 PBS. The paddle rotation speed was kept at 50 rpm and temperature was kept constant at 37 ± 0.5 °C throughout the process. The 5 mL of sample was drawn at each interval (0, 0.5, 1, 2, 4, 6, 8, 10, 14, 18, 22 and 24 h) and filtered for analysis. To maintain sink conditions of the formulation discs, 5 mL of fresh dissolution was also replaced after each sample withdrawal. The following formula was used to estimate the percentage of insulin release:$$Drug\;release\;\left(\%\right)=\frac{F_{t}}{F_{load}}\times 100$$

In this equation, *F_t_* shows the insulin quantity released at time *t* and *F_load_* denotes the amount of insulin loaded in the BA hydrogel and hydrogel NC [32].

### 2.14. Application of Kinetic Models of Drug Release

To find and compare the best fitted kinetic model and insulin release mechanism, all BA-hydrogel and hydrogel-NC formulations’ release data were subjected to analysis using a Microsoft Excel^®^ Add-Ins program named DD-Solver. Kinetic models applied were zero order, first order, Higuchi, Korsmeyer-Peppas and Hixson–Crowell models. For the first model of insulin release, the equations used was,
*M_t_* = *K*_1_ × *t*

For the zero order release model, the kinetic equation implied was,
*M_t_* = *K*_0_ × *t*

Here, *M_t_* is the insulin release quantity at any time *t. K*_1_ is the first order rate constant. Meanwhile, *K*_0_ is the zero order rate constant. The insulin release mechanism was also determined by applying the Higuchi model using the equation,
*M_t_*/*M*_0_ = *K_H_* × *t*_1/2_

Here, *K_H_* is the Higuchi rate constant. The mechanism of insulin release was also checked by applying the Korsemeyer-Peppas model with the equation,
*M_t_*/*M_∞_* = *Kt^n^*

In this equation, *M_t_/M_∞_* denote the quantity of insulin released at any time interval *t*. The “*K*” is the rate constant of insulin release and “n” is the release exponent showing the mechanism of drug release. If the value of n = 0.89 or >0.89 then it presents case II transport and super case II transport, respectively. The value of n = 0.45 denotes Fickian diffusion, and a value in the range of 0.45–0.89 shows non-Fickian anomalous diffusion [33]. The equation for the Hixson–Crowell model is
(1 − *M_t_*/*M_∞_*)^1/3^ = 1 − *Kt*

Here, *M_∞_* is the maximum amount of drug release at the maximum time interval, while *K* is the rate constant [34].

### 2.15. Acute Oral Toxicity Studies

These studies were conducted for 14 days for analysing effect of Ins-Mmt-BA hydrogel NCs on the physical activity and biochemistry of blood and vital organs of the albino rabbits. For this study, the Organization for Economic Cooperation and Development (OECD) guidelines were strictly followed. The whole study was conducted under the permission of the University of Lahore, Institutional Research and Ethics Committee (IREC). The IREC-allotted vide notification number was IREC-2020-22. The experimental animals were placed into two groups, control (Group A) and treated (Group B), each with six rabbits (n=6). A total of 12 healthy albino rabbits were kept in the University of Lahore animal house in stainless steel cages for 1 week to acclimatize them. Before initiating the experiment, the animals were exposed to a cycle of day and night of 12 h duration each and also fasted overnight. Then, the hydrogel NC discs were crushed to powder form and a calculated dose was administered to Group B. All the animals were carefully monitored for any changes in their physical activity, body weight, food and water intake, skin allergies and primary eye irritation. Samples for complete blood count (CBC), lipid profile, renal and liver profiles [aspartate aminotransferase (AST) and alanine aminotransferase (ALT)] were taken at the 7th and 14th days of the experiment. At the 14th day after giving the dose, all the animals were sacrificed by giving anaesthesia (Ketamine/Xylazine; 70/30). The vital organs such as the heart, liver, kidney, small intestine, lungs and spleen were washed, weighed and stored in 10% formalin solution in air-tight containers separately for histopathology studies. The organs were sliced carefully and observed under an optical microscope at different resolutions after staining with Hematoxylin and Eosin (H & E) [35].

### 2.16. In Vivo Pharmacodynamics Parameters Evaluation

A crossover study was designed to evaluate the insulin intestinal absorption and its release pattern over time from the developed BA hydrogels versus Ins-Mmt-BA hydrogel NCs. For this purpose, under the approval of IREC vide notification number IREC-2020-22, strictly following OECD guidelines, a total of 30 healthy male albino rabbits (≥2 kg weight) were taken. All animals were exposed to 12 h light and darkness cycles for 7 days to acclimatize with ambient atmosphere. For diabetes induction, all rabbits were administered alloxan monohydrate injection (Sigma-Aldrich, Germany) in a single intraperitoneal dose of 150 mg/kg. A 5% glucose solution was administered ad libitum for 24 h to avoid alloxan-induced hypoglycemia. To confirm diabetes induction all, animals’ blood was withdrawn daily from the ear marginal vein for checking blood glucose levels with the help of a glucometer and glucose strips (ACCU-CHEK^®^, Roche, Basel, Switzerland) for 10 days. Rabbits displaying fasting blood glucose levels higher than 150 mg/dL were considered as diabetic [36].

After diabetes induction, the rabbits were placed into 5 groups (n = 6) randomly. The first group was orally administered with ground Ins-BA hydrogel (10 IU/kg) by an oral gavage needle with 5 mL of water to ensure complete dose administration. The second group was given orally the ground Ins-Mmt-BA hydrogel NC (10 IU/kg). The third group, acting as a positive control, was subcutaneously (SC) injected with standard Humulin-R^®^ insulin (5 IU/kg). The fourth group was orally administered with insulin solution (10 IU/kg). The fifth group, acting as a negative control was administered with blank hydrogel. Blood samples were taken at specified intervals (0, 1, 2, 3, 4, 6, 8, 10, 12, 16, 20, 22 and 24 h) from the jugular vein and collected in EDTA containing sample collection tubes. The blood samples were then centrifuged at 3500 rpm for 15 min and separated plasma samples were stored at –20 °C for further detection of insulin by the HPLC method. At the same intervals, blood samples were also checked for blood glucose levels by a glucometer for observing the pharmacodynamic effect of the administered dosage forms [8].

### 2.17. HPLC Method Development and Pharmacokinetic Evaluation

For pharmacokinetic evaluation, a previously developed HPLC method for insulin detection in plasma was used with slight modifications [37]. The HPLC system (Hitachi, Tokyo, Japan) comprising a Gilson delivery pump with a 6-valved sample injection port fitted with a 20 µL sample loop, A UV-visible detector, thermostat column and chromato-integrator were used for sample analysis. The RP-C_18_ (150 × 4.60 mm ID, 5 µm) analytical column (Phenomenex, Torrance, CA, USA) was used. The mobile phase was prepared by 0.2 M Na_2_SO_4_ solution with pH adjusted to 2.3 with orthophosphoric acid mixed with acetonitrile (74:26) and filtered through 0.5 µm membrane filters and sonicated for 10 min to remove dissolved gas bubbles. The stock solution of insulin at 140 µg/mL strength was made in PBS. The serial dilutions of insulin solution (3.5 µg/mL to 140 µg/mL) after filtering through a 0.2 µm membrane filter were injected into a chromatographic column to find the linearity curve to develop a method for insulin detection. The linearity curve with the same serial dilutions of insulin in plasma was also determined. For sample preparation from collected plasma, 200 µL of thawed plasma (collected at different intervals) was mixed with 800 µL of n-hexane:dichloromethane (1:1). The mixture was centrifuged for 10 min at 3500 rpm (Hitachi Zentrifugen EBA 20, Hitachi Ltd., Tokyo, Japan). The supernatant with extracted insulin was used after filtering through a 0.22 µm membrane filter (Sartorius, Göttingen, Germany) for injection into the HPLC system. All samples were injected into the chromatographic column at 25 °C, with 1 mL/min mobile phase flow rate for 10 min, and t λ_max_ was kept at 214 nm. Later on, a Microsoft Excel^®^ Add-Ins program named pK-solver was used for pharmacokinetic (pK) parameters evaluation from the HPLC collected data [35,38]. After this, the percentage of relative bioavailability (% RBA) was calculated for the developed hydrogel and hydrogel NC formulations from these pK parameters by the following formula.
$$\%RBA=\frac{\left(AUC\right)oral\times Dose\;s.c}{\left(AUC\right)s.c\times Dose\;oral}\times 100$$

Here, (*AUC*)oral and *(AUC)s.c* are the areas under curves of our developed oral insulin formulation and *S.C* standard, respectively. Meanwhile, *Dose oral* and *Dose s.c* are the administered doses of developed oral insulin and S.C standard insulin formulations.

### 2.18. In Vitro Biodegradation Studies

Biodegradation studies were conducted on optimized formulations of hydrogel and hydrogel nanocomposites under simulated gastric and intestinal conditions. The same approach was used as was adapted by Batool et al. (2022) with little variations. Three litters of acidic (HCl) and alkaline (phosphate) buffers were prepared as per USP monograph, separately. A precise quantity of pepsin (~3 g/900 mL) and pancreatin (9 g/900 mL) was separately added into acidic and basic buffer, respectively, in order to make respective simulated solutions. Both media were transferred into baskets of dissolution USP Type II apparatus. Hydrogel discs (known weight) of freshly prepared networks after complete washing were poured into these media. Experimental temperature was set at 37 °C with a paddle speed of 50 rpm. Discs were recovered at pre-set time intervals for weight loss determination with respect to time after blotting the surface with filter paper [39].

### 2.19. Statistical Data Analysis

GraphPad Prism version 5.01 was employed for all types of statistical analysis. The unpaired Student’s t-test was employed for detecting significant differences between two independent groups. If the calculated * *p* ≤ 0.05, then it represents that the null hypothesis H_o_ is rejected and there is a significant difference present between two groups. Meanwhile, ** *p* ≤ 0.01 or *** *p* ≤ 0.001 imply that the possibility of wrong rejection of H_o_ are even minor. Results were displayed as mean ± standard deviation (SD) for at least three trials.

## 3. Results and Discussion

### 3.1. Preparation of Black Seed Polysaccharide Extract (BSPE)

Black seed polysaccharide extract (BSPE) was effectively prepared. A brownish black, sticky and water-soluble extract was obtained. Percentage yield of the extract was 4.97%. The yield was closer to the work reported by Trigui et al. in 2018 with 5.18% yield [19].

### 3.2. Preparation of Insulin-Loaded BA Hydrogels and Ins-Mmt-BA Hydrogel NCs

The six formulations of BA hydrogels were developed (BA1-BA6) by varying concentrations of the crosslinker (MBA) and monomer (AA). The contents of all formulations are given in Table 1. From BA1-BA3 hydrogel formulations, structural smoothness and elasticity was decreased with an increase in MBA content. Meanwhile, from BA4-BA6 hydrogels formulations, both the elasticity and structural smoothness were enhanced with the increase in AA content.

Another six formulations of BA hydrogel NCs (Ins-Mmt-BA1 to Ins-Mmt-BA6) were also efficiently developed by post-loading of Ins-Mmt to optimally swollen BA hydrogels (BA1-BA6) in pH 7.4 PBS (with maximum hydrogel pore openings for instant loading of Ins-Mmt to BA hydrogels, avoiding insulin release in the PBS). All hydrogel NCs seemed to be less elastic compared to their respective hydrogels. The proposed structure of both hydrogel and hydrogel NCs is shown in Figure 1.

### 3.3. Swelling Studies

Swelling studies such as swelling ratio (q) were conducted on each formulation of hydrogel (BA1-BA6) and hydrogel NC (Ins-Mmt-BA1 to Ins-Mmt-BA6) at pH 1.2 and pH 7.4. The impact of pH and hydrogel constituents (crosslinker and monomer) was observed on the hydrogel swelling ratio (Figure 2a,b). The effect of pH and loaded Ins-Mmt on hydrogel NC swelling ratios was also observed (Figure 2c,d)

It can be observed (Figure 2a,b) that all hydrogel (BA1-BA6) preparations showed a negligible maximum swelling ratio at pH 1.2 (<2.6) compared to pH 7.4 (>20.18) at 72 h. Similarly, in all hydrogel NC (Ins-Mmt-BA1 to Ins-Mmt-BA6) formulations (Figure 2c,d), again a negligible maximum swelling ratio at pH 1.2 (<2.54) compared to pH 7.4 (>18.92) at 72 h was observed. Hence, both BA hydrogel and hydrogel NCs showed pH responsiveness (*p*-value < 0.001). Thus, a highly significant difference was shown between swelling of all formulations at pH 1.2 compared to pH 7.4. The reason was the presence of more charged carboxylic groups (-COO^−^) at pH 7.4 as compared to pH 1.2 in both hydrogels and hydrogel NCs, resulting in greater repulsive forces and creating greater pore openings and uptake of more swelling media in respective networks, resulting in more swelling. Meanwhile, at pH 1.2, there is less availability of charged carboxylic groups (-COO^−^), resulting in less repulsive forces, less pore opening and restricted water entry, thereby causing less swelling [35].

In BA1-BA3 hydrogel preparations (Figure 2a), a decreasing trend in swelling ratio (31.72–20.18) at pH 7.4 was observed with the increase in crosslinker (MBA) contents (*p*-value < 0.01). This is due to increased crosslinking density with increasing MBA contents in hydrogel networks because of more crosslink points created by MBA. That resulted in lesser expansion of hydrogel networks and, hence, decreased swelling [34]. Similarly, in BA4-BA6 hydrogel formulations (Figure 2b), the increase in swelling ratio (29.38–55.55) at pH 7.4 was seen with an increase in the amount of monomer (AA) (*p*-value < 0.01). This increase is obviously because of the presence of more charged carboxylic groups (-COO^−^) of AA, resulting in greater repulsive forces and, hence, expansion of polymeric chains at higher pH, more swelling media intake and more resultant swelling of hydrogel networks [38]. The *p*-value showed that both MBA and AA contents significantly affected the swelling of hydrogel formulations.

As far as the effect of post-loaded Ins-Mmt on hydrogel NC swelling is concerned, it can be clearly seen (Figure 2c,d) that, in comparison to all hydrogel (BA1-BA6) formulations, the hydrogel NCs (Ins-Mmt-BA1 to Ins-Mmt-BA6) showed a restricted swelling over time. From Ins-Mmt-BA1 to Ins-Mmt-BA3, the swelling ratio was in the range of 30.12–18.92 (*p*-value < 0.05). From Ins-Mmt-BA4 to Ins-Mmt-BA6 it was in the 27.02 to 52.34 range (*p*-value < 0.05), showing that the impact of Ins-Mmt contents on hydrogel NC swelling was also significant. The presence of crosslinked Ins-Mmt in hydrogel NCs resulted in more and more crosslink density with resultant restricted swelling compared to respective hydrogels [23].

### 3.4. Drug Loading Determination and Their Inter-Relationship with Swelling

For the loading of insulin into BA hydrogels (BA1 to BA6), all formulation discs were swollen to equilibrium in a sufficient quantity of pH 7.4 PBS containing 60 IU of insulin, thus formulating direct insulin-loaded BA hydrogels. BA1-BA6 hydrogel formulations were also loaded with Ins-Mmt containing 60 IU of intercalated insulin to prepare respective hydrogel NCs (Ins-Mmt-BA1 to Ins-Mmt-BA6). DLE% was determined for all developed preparations and was recorded in Table 2. A direct inter-relationship was observed between maximum swelling ratio (q) and DLE% of all developed hydrogels and respective NCs, as can be seen in Table 2. From BA1-BA3, the maximum q decreased from 31.72 to 20.18, so the DLE% decreased from 86.67% to 70.45%. This is definitely due to the fact that as the swelling of hydrogel decreased with increasing crosslink density, less drug solution entered into the hydrogel network, causing less drug loading. Conversely, in BA4-BA6 hydrogels, as the maximum q arose from 29.38 to 55.55, the DLE% rose from 72.58% to 90.33%. The reason for the increase in DLE% is again clear that as more drug solution enters the hydrogel network with increase swelling, more drug is entrapped in hydrogels [26].

In comparison, in the case of hydrogel NCs, from Ins-Mmt-BA1 to Ins-Mmt-BA3 formulations, again the DLE% was decreased from 81.24% to 66.67%, with a decrease in maximum q from 30.12 to 18.92. However, DLE% showed an increasing trend in the case of Ins-Mmt-BA4 to Ins-Mmt-BA6 hydrogel NCs, from 68.11% to 85.45% with a rise in maximum q from 27.02 to 52.34. It can be clearly observed in Table 2 that an overall decrease in both maximum q and resultant DLE% occurred in hydrogel NCs compared to respective hydrogel formulations. This effect is because of the presence of Ins-Mmt in hydrogel NCs, causing a decrease in maximum q with an increase in crosslink density and, hence, a relative decrease in overall DLE% [40].

### 3.5. Sol-Gel Analysis of BA Hydrogels versus Ins-Mmt-BA Hydrogel NCs

Gel fraction was determined for all developed hydrogel (BA1-BA6) and hydrogel NC (Ins-Mmt-BA1 to Ins-Mmt-BA6) formulations to find out and compare the percentage of crosslinked polymer. It can be observed in Figure 2 that, from BA1-BA3 hydrogel preparations, the gel fraction was increased from 90.59% to 95.49% with the increase in crosslinker content. Similarly, from BA4-BA6 hydrogel formulations, again an increase in gel fraction was seen, from 87.44% to 98.80%, with the increase in monomer contents. This increase in gel fraction with the increase in both monomer and crosslinker contents is due to more crosslink points created in hydrogel networks with more monomer and crosslinker concentrations [40].

In comparison, in the case of hydrogel NC formulations, the same increasing trend in gel fraction can be visualized in Figure 3 from Ins-Mmt-BA1 to Ins-Mmt-BA3 (91.58–96.33%) and from Ins-Mmt-BA4 to Ins-Mmt-BA6 (88.31–99.67%). It was again due to the same reason of increased number of crosslink points with more MBA (crosslinker) and AA (monomer) contents, respectively [30,39]. However, the comparison in Figure 3 also depicts a relative increase in gel fraction in all hydrogel NC (Ins-Mmt-BA1 to Ins-Mmt-BA6) formulations compared to all hydrogel (BA1-BA6) preparations. The obvious reason for this is the loaded Ins-Mmt in hydrogel NCs with its added crosslinking points [41].

### 3.6. FTIR Analysis

FTIR spectra of BSPE, AA, BA hydrogel, Mmt-Na^+^, Ins-Mmt, Ins-BA hydrogel and Ins-Mmt-BA hydrogel NC were recorded (Figure 4). The spectrum of BSPE showed a broad peak at 3355 cm^−1^, representing the stretching vibrations of the hydroxyl group (-OH) of inter- and intra-molecular H-bands. A relatively smaller band, at 2943 cm^−1^, was attributed to the bending and stretching vibrations of the aliphatic (C-H) group. A strong peak, observed at 1654 cm^−1^, presented the asymmetric stretching vibrations of the carboxylate (COO^−^) group of uronic acid. Another small band at 1419 cm^−1^ was ascribed to the symmetric stretching mode of the carboxylate (COO^−^) group. Above, two peaks confirmed the presence of galacturonic acid and glucuronic acid in the polymer extract of black seed. A very strong peak at 1037 cm^−1^ was attributed to the pyranose form of sugars [19].

As far as the FTIR spectrum of AA is concerned, the wide band at 2900–3300 cm^−1^ present hydroxyl (O-H) groups. The spectrum also showed two quite distinctive peaks at 1697 cm^−1^ and 1615 cm^−1^, showing correspondence to carboxylic (C=O) and C=C stretching, respectively.

The spectrum of unloaded BA hydrogel showed a prominent peak at 1447 cm^−1^, corresponding to CH_2_ bending vibrations. Moreover, the peak at 1213 cm^−1^ can be assigned to C-O stretching. If we compare the AA and BA hydrogel spectra, then the disappearance of the peak at 1615 cm^−1^ (related to C=C stretching) in BA hydrogel confirms successful crosslinking, as MBA crosslinks randomly to a C=C moiety on BA hydrogel, forming C-C linkages [42].

In Ins-BA spectra, it can be clearly seen that all other characteristic bands of BA hydrogel are present along with two additional peaks at 1652 cm^−1^ and 1539 cm^−1^, representing amide I and amide II bands present in the insulin structure. The presence of these two peaks confirms the successful loading of insulin in BA hydrogels with no significant denaturation [10].

The spectrum of Mmt-Na^+^ clay presented a small, sharp absorption band at 3620 cm^−1^, related to the stretching vibrations of the free -OH groups. A relatively wide band at 3372cm^−1^ was because of the stretching vibrations of –NH of NH^4+^ ions existing in the inter-layers of Mmt-Na^+^. A sharp peak at 1633 cm^−1^ was attributed to the bending vibrations of the OH group of the interlayer water molecules present in the clay or lattice water. Different small, overlapped or sharp absorption bands with different intensities were also observed around 1080–1000 cm^−1^ (maximum at 1027 cm^−1^), and 915 and 752 cm^−1^ were attributed to the stretching vibrations of the in-plane Si–O and bending vibrations of -OH of Me–OH groups (primarily AlAlOH, AlMgOH, AlFeOH present in the structure of Mmt-Na^+^) seen in the spectrum, respectively [43]. The spectrum of Ins-Mmt represents the characteristic peak at 3370 cm^−1^, related to the stretching vibrations of NH and OH bonds of insulin. Meanwhile, absorption peaks at 1653 and 1551 cm^−1^ are attributed to amide I and II bands, respectively, in the insulin structure. The presence of these characteristic bands of insulin in this spectrum can confirm the successful loading of insulin in the clay with no significant denaturation [8,44].

In the FTIR spectrum of Ins-Mmt-BA hydrogel NCs, the absorption peaks of amide I and II at 1652 cm^−1^ and 1538 cm^−1^, respectively, confirms the successful insulin loading into hydrogel NCs with no significant denaturation. Meanwhile, merged absorption bands with different intensities at 993 cm^−1^ and 921 cm^−1^ confirm the presence of loaded Mmt clay in the hydrogel NCs [10,45].

### 3.7. Scanning Electron Microscopy (SEM)

The surface morphology of unloaded BA hydrogels and Ins-Mmt-BA hydrogel NCs at different magnification powers was determined via SEM analysis for the formulations showing optimum swelling. The surface morphology and presence of pores in both hydrogels and hydrogel NCs has important effects on the entry of the solvent and the rate of drug release from the developed formulations. SEM photomicrographs are given in Figure 5. It can be observed in Figure 5a,b that, in the unloaded BA-hydrogel surface, the pores or voids are widely distributed on a relatively rough surface. Meanwhile, in the case of Ins-Mmt-BA hydrogel NCs in Figure 5c,d, these pores mostly seemed to be filled with the loaded Ins-Mmt clay, thus, confirming proper loading of Ins-Mmt clay in BA hydrogels to form respective hydrogel NCs [46].

### 3.8. Energy Dispersive X-ray Spectroscopic Analysis (EDS)

The comparative analysis of the EDS spectrum of unloaded BA-hydrogel, Mmt-Na^+^ clay and Ins-Mmt-BA hydrogel NC was performed using EDS spectroscopy (Figure 6) to confirm the loading of Ins-Mmt clay into the BA hydrogel to form an Ins-Mmt-BA hydrogel NC. It can be observed (Table 3) that the elemental composition of unloaded BA hydrogel was C (57.32%), N (9.14%), O (33.23%) and K (0.31%). While for Mmt-Na^+^, the elemental composition was O (55.81%), Mg (0.98%), Al (6.18%), Na (1.08%), Si (34.73%), K (0.64%), Ca (0.20%), Ti (0.25%) and Fe (1.22%). In the case of Ins-Mmt-BA hydrogel NCs, the elemental composition recorded was C (36.27%), N (8.88%), O (28.94%), Na (7.99%), Mg (0.27%), Al (1.43%), Si (3.99%), P (10.31%), S (0.35%), K (0.26%) and Ca (1.31%). The presence of all main elements of Mmt-Na^+^ clay, such as Na, Mg, Al, Si and Ca, in the Ins-Mmt-BA hydrogel NC spectrum confirmed its successful loading into BA hydrogels to form hydrogel NCs. Moreover, the presence of element S also confirmed the successful loading of insulin in hydrogel NC formulation [47].

### 3.9. Powdered X-ray Diffraction (PXRD) Analysis

X-ray diffraction analysis of Ins-Mmt clay, unloaded BA hydrogels, Ins-BA hydrogels and Ins-Mmt-BA hydrogel NCs was performed to confirm and compare the changes in the physical form of insulin in hydrogel and hydrogel NC formulation. It can be seen in Figure 7 that, in case of the Ins-BA hydrogel diffractogram, similar types of patterns and peaks were observed as those of unloaded BA hydrogel. However, in the Ins-BA hydrogel diffractogram, from 5° to 10° angles there was a rise in intensity of peaks, and then from 15° to 25° angles a broad peak was observed. Bueche had worked on the stability of PEG-insulin and insulin hexamer assemblies in solution and dry powder state. The X-ray diffractogram of pure insulin crystals can be observed in his thesis with similar peaks. Therefore, it was confirmed that, in the case of Ins-BA hydrogels, the insulin was successfully loaded with its intact crystalline hexameric structure. As far as the XRD pattern of Ins-Mmt-BA hydrogel NCs is concerned, it has all the characteristic peaks of a BA hydrogel and Ins-Mmt with midway intensities, confirming successful loading of Ins-Mmt into BA hydrogels to form NCs. In addition, at the similar angles of 5° to 10° and from 15° to 25°, the XRD pattern of Ins-Mmt-BA hydrogel NCs confirmed the successful insulin loading with the intact crystalline hexameric form [48,49].

### 3.10. Thermal Studies/Differential Scanning Calorimetry (DSC) and Thermogravimetric Analysis (TGA)

Thermal changes such as enthalpy changes and phase transition temperature of unloaded BA hydrogel and thermal stability of BA hydrogel versus Ins-Mmt-BA hydrogel NCs were assessed by DSC and TGA studies.

The DSC thermogram of unloaded BA hydrogel (Figure 8a) represented an endothermic peak at 287.4 °C, presenting hydrogel decarboxylation. The DSC curve of Ins-BA hydrogel (Figure 8b) showed a prominent endothermic peak at 224.75 °C with an enthalpy change (ΔH) of −33.679 J/g, corresponding to oxidative degradation of the Ins-BA hydrogel system. Meanwhile, the DSC thermogram of an Ins-Mmt-BA hydrogel NC (Figure 8c) showed an endothermic peak at 216 °C, corresponding to hydrogel NC decomposition with an enthalpy change (ΔH) of −23.961 J/g. According to the literature, the endothermic peaks in DSC thermograms occurring at a very early stage indicate moisture loss. Meanwhile, at a later stage these peaks either represent melting points or oxidative degradation [32,50].

In the case of the TGA thermogram of unloaded BA hydrogel (Figure 8a), only a weight loss of 0.7% compared to initial sample weight was observed at 100 °C, due to moisture loss. A weight loss of 8.18% was detected at 224 °C, representing oxidative degradation of the hydrogel. Moreover, the 51.05% weight loss was seen at 337.81 °C, presenting the first stage of decomposition mainly due to hydrogel decarboxylation. From 338 °C to 500 °C, a further decrease in weight until 84.5% was recorded, showing the second stage of sample decomposition. Within the TGA thermogram of the Ins-BA hydrogel (Figure 8b), a weight loss of 4.16% was seen at 100 °C due to the vaporization of entrapped water molecules. A further decrease in weight of 21.64% was observed at 220.28 °C, corresponding to oxidative degradation of the hydrogel. From 221 °C to 390.13 °C, a total weight loss of 47.89% was recorded, presenting the first stage of sample decomposition. Meanwhile, from 391 °C to 500 °C, weight loss until 66.62% was detected due to the second stage of decomposition of the hydrogel sample. In comparison, the TGA thermogram of the Ins-Mmt-BA hydrogel NC (Figure 8c) showed a completely different pattern, with a weight loss of 2.7% at 100 °C due to moisture loss. Until 210.57 °C, this weight loss was increased only up to a total of 14.26%. Meanwhile, from 211 °C to 217.98 °C, an abrupt weight loss was observed until 88.76% due to the first stage of hydrogel NC decomposition. From 218 °C to 500 °C, a further decrease in weight until 98.99% was recorded, depicting the second stage of decomposition of the hydrogel NC. Kevadiya et al. have reported similar TGA thermograms for Vitamin B1 or Vitamin B6-MMT-alginate nanocomposites, with starting sample decomposition stages from 200 °C [51]. As far as the comparison of thermal stability is concerned, up to 200 °C it can be observed from the above results that less weight loss was observed for hydrogel NCs compared to hydrogel formulations at 100 °C (2.7% for NC, 4.16% for gel). Similarly, at 200 °C, 12.97% weight loss for NC and 16.40% for hydrogel clearly indicate that, until 200 °C, the Ins-Mmt-BA hydrogel NC formulation showed more thermal stability compared to the Ins-BA hydrogel preparation.

### 3.11. In Vitro Release Studies

Insulin release against time was determined from all hydrogel (Ins-BA1 to Ins-BA6) and hydrogel NC (Ins-Mmt-BA1 to Ins-Mmt-BA6) formulations. In vitro release profiles of Ins-BA hydrogels with different MBA (Ins-BA1 to Ins-BA3) and monomer AA (Ins-BA4 to Ins-BA6) concentrations are given in Figure 9a. It can be observed that, in Ins-BA1 to Ins-BA6, the percentage release of insulin was less than 2.3% for the first 2 h at pH 1.2. After that, up to 12 h, the percentage release was dramatically increased at pH 7.4 in all hydrogel formulations. In Ins-BA1 to Ins-BA3 hydrogels, the drug release was decreased from 96.08 to 86.17% with an increase in MBA contents (*p*-value > 0.05). This effect was due to an increase in crosslink density with the increase in MBA contents and resultant decreased swelling of hydrogel networks, leading towards a decrease in drug percentage release too. In the case of Ins-BA4 to Ins-BA6 hydrogels, the maximum release was increased from 91.14% to 99.38% with an increase in AA contents (*p*-value > 0.05). This increase in % release was due to more swelling of the hydrogel network because of greater repulsive forces caused by a larger number of ionized carboxylic acid (-COOH) groups of AA at pH 7.4 [25,29]. A *p*-value > 0.05 indicated that no significant difference existed between release patterns of insulin from hydrogel formulations.

Meanwhile, in the case of Ins-Mmt-BA1 to Ins-Mmt-BA6 hydrogel NCs (Figure 9b), the same trend was observed in drug release. For example, by increasing MBA content in Ins-Mmt-BA1 to Ins-Mmt-BA3, a decrease in % release from 93.52% to 82.01% was observed (*p*-value > 0.05). Similarly, by increasing AA contents in Ins-Mmt-BA4 to Ins-Mmt-BA6, an increase in swelling and % release was observed from 88.14% to 96.17% (*p*-value > 0.05). Thus, it is also evident that no significant difference between release patterns of insulin from hydrogel NC formulations was present. However, an overall increase in release time (up to 24 h) for all hydrogel NC formulations can be observed as compared to hydrogels (up to 12 h). This difference in insulin release time is due to incorporation of Ins-Mmt in all hydrogel NC formulations, causing an increase in crosslink density. This resulted in slower swelling of the hydrogel NC network, hence, an increase in release time [10].

This increase in insulin release time could have significant effects on the in vivo protection of insulin from enzymatic degradation and bioavailability. If a formulation manages to control insulin release in the gastric and small intestinal environments and releases most of the insulin in the large intestine (where the level of brush-border and luminal proteases is very low compared to the duodenum and jejunum), there are chances of less insulin degradation and improved bioavailability [6].

### 3.12. Kinetic Modelling of Release Data

To access the drug release kinetics, different kinetic models were applied on in vitro release data with the help of an MS Excel-based Add-Ins software named “DD Solver”. Zero order, first order, Higuchi, Korsemeyer-Peppas and Hixson–Crowell kinetic models were applied on release data. According to their data of regression coefficient (R^2^) and release rates (Appendix A), the best fit model followed by all developed formulations was the Korsemeyer-Peppas model with (R^2^ = 0.9752 to 0.9936). The values of exponent “*n*” for all hydrogel and hydrogel NC formulations were >0.89, indicating a super case II transport mechanism of insulin release. This means that drug release was affected by diffusion and relaxation of polymeric chains within the network [52,53].

### 3.13. Acute Oral Toxicity Studies

Acute oral toxicity studies were conducted for the Ins-Mmt-BA6 hydrogel NC formulation, which was selected after completion of in vitro studies. Toxicity studies were conducted on 12 healthy albino rabbits divided into 2 groups (n = 6), 1 was selected as control (C) and other as treated (T). The studies were continued for 14 days from administration of the experimental formulation. The animals were subjected to physical, haematological, biochemical and histological examinations. During physical observations, rabbits were investigated for body weight, food and water intake and any ocular or dermatological irritation at the 1st, 7th and 14th days of administration of experimental formulation (Table 4). No prominent change in body weight or food and water intake was noted. In addition, no ocular or dermatological irritation or death of any animal was observed. During haematological and biochemical investigations, complete blood count (CBC), hepatic and renal function tests and lipid profiles were investigated. According to CBC results, there was no sign of infection or significant difference from normal values of both groups noted (Table 5). The hepatic and renal function tests and lipid profiles were also normal for both groups, indicating no sign of fatty liver disease, renal damage or alteration of triglyceride levels (Table 6). After completion of 14 days of the experiment, all animals were slaughtered and major organs were taken out, weighed, washed and stored in 10% formalin solution for histological observations. The major organs (heart, liver, kidney, lungs, intestine and spleen) were then sliced for making stained slides by H and E stains. Slides were observed at different magnifications, and no inflammation, lesions or degeneration was seen in any slide (Figure 10). Moreover, no hypoxia, cardiomegaly or hypertonicity of cardiomyocytes was detected for both groups. The lung tissues of both groups showed no signs of haemorrhage, with normal alveolar wall thickness, and there was no sign of fluid accumulation in the lumen of alveoli. Liver tissues also showed no fibrosis or accumulation of immune cells or extracellular fluid for both treated and control groups. Spleen tissue samples of both groups also showed no splenomegaly and red and white pulp zones were properly distinguishable with uniform white blood cell distribution in the white area. The intestinal slides of both treated and control groups showed good gross morphology, no signs of ulceration, epithelial cells were covered with secreted protein and alveoli showed no hyperplasia. Kidney slides also showed no fibrosis or immune infiltration, and the bowman capsules and glomerulus were intact for both groups. Therefore, these results confirm that the developed black seed hydrogel NC formulation did not show any acute toxicity profile.

### 3.14. In Vivo Pharmacodynamics Parameters Evaluation

For checking the decrease in blood sugar levels of diabetes-induced experimental rabbits by the prepared formulations, a crossover study was designed. The 30 rabbits were divided into 5 groups (n = 6). Four groups were orally administered with blank hydrogel (negative control group), Ins-BA hydrogel, Ins-Mmt-BA hydrogel NC and oral insulin solution with the same strength of insulin (10 IU/kg). The fifth group was administred with SC standard insulin (5 IU/kg). The blood samples from each group were taken at different intervals for 24 h and a graph was drawn for blood glucose level (% of initial) versus time (Figure 11a). It can be seen that the group administered with blank hydrogel did not show any decrease in percentage blood glucose levels (% BGL). Almost similar results were shown by the group administed with oral insulin solution, with no significant decrease in % BGL. The reason is the destruction of insulin in the gastric environment due to gastric enzymes [6]. As far as the SC standard insulin-administered group is concerned, it showed a maximum decrease in % BGL up to 13.55% within the first 3 h, but the % BGL rose to 59.67% at the 8 h interval. The group given oral Ins-BA hydrogel showed a maximum decrease in % BGL (64.84%) at 8 h, and these levels rose to 78.57% at the 12 h interval. Lastly, in the case of the Ins-Mmt-BA hydrogel NC-treated group, the % BGL was decreased to a maximum of 52.61% at the 20 h interval and rose up to only 66.52% at the 24 h interval. Thus, it is evident that oral Ins-Mmt-BA hydrogel NC formulation showed a more prolonged and sustained decrease in % BGL of diabetic rabbits as compared to oral Ins-BA hydrogel formulation, oral insulin solution and SC standard insulin injection. This effect is definetly due to its restricted insulin release in the stomach and slow and gradual release of insulin in the intestinal environment [10,54].

### 3.15. Pharmacokinetic Evaluation

To further find out and compare the pK parameters and bioavailability of the developed hydrogel and hydrogel NC formulaions to that of SC standard insulin formulations, insulin plasma levels were also estimated by using a previously developed HPLC method with slight modifications [35]. The chromatogram of plasma with insulin showed the retention time of insulin at 8.487 min (Figure 12).

The plasma samples collected from all rabbit groups at different time intervals were investigated for their concentrations. A graph was constructed for plasma insulin concentration versus time (Figure 11b). pK-Solver (Microsoft Excel^®^ Add-Ins program) was used on the data obtained to find out pK parameters (Table 7) such as maximum plasma concentration (C_max_), time for C_max_ (t_max_), plasma half life (t_1/2_), mean plasma residence time (MRT), area under the curve (AUC)_0–∞,_ volume of distribution (V_d_) and clearance (Cl). The C_max_ for SC insulin standard was 1.735 µg/mL and t_max_ was 3 h. It had a plasma t_1/2_ of 3.493 h. Meanwhile, MRT for SC standard was 6.007 h, (AUC)_0–∞_ was 12.985 µg h/mL and there was a Cl of 13.361 mL/h. As far as the comparison of these pK parameters with respect to the developed hydrogel and hydrogel NC formulations is concerned, oral Ins-BA hydrogel showed a C_max_ of 0.288 µg/mL at a t_max_ of 8 h. However, for the oral Ins-Mmt-BA hydrogel NC, the C_max_ and t_max_ were higher in comparison to 0.347 µg/mL and 20 h, respectively. Similarly, the half life (t_1/2_) of the hydrogel formulation was 1.065 h, compared to that of hydrogel NC at 20 h. This is because of the slow release of insulin from the hydrogel NC over the passage of the intestine, allowing more time for insulin absorption. It can be easily observed (Table 7) that MRT for hydrogel preparation was much less compared to hydrogel NC (9.608 h and 24.390 h, respectively). The (AUC)_0–∞_ was also great for Ins-Mmt-BA hydrogel NC in comparison to the respective hydrogel that was 6.831 µg h/mL to 2.616 µg h/mL. Lastly, the clearance (Cl) for the Ins-BA hydrogel was higher, at 132.630 mL/h, compared to the Ins-Mmt-BA hydrogel NC, at 50.799 mL/h. Meanwhile, the volume of distribution was less for the Ins-BA hydrogel, at 203.83 mL, as compared to the Ins-Mmt-BA hydrogel NC, for which V_d_ was 584.66. The difference in these pK parameters is again due to the slow and prolonged release of insulin from the hydrogel NC formulation [55]. The % RBA for the developed BA6 hydrogel was 10.07%. Moreover, the % RBA for the developed Ins-Mmt-BA6 hydrogel NC formulation was 26.3%. That is much higher than some previously developed hydrogel systems for oral insulin delivery [54,55,56], thus making the Ins-Mmt-BA hydrogel NC a favourable system for safe and prolonged oral delivery of insulin. Moreover, the increase in insulin release time in the hydrogel NC has definitely showed a significant impact on oral bioavailabilty [6].

### 3.16. In Vitro Biodegradtion Studies

Optimized polymeric combinations have presented variable degradation responses at variable pH of 1.2 and 7.4. In a 12 hr study, BA6 hydrogels exhibited 72% weight loss, while, in comparison to this, Ins-Mmt-BA6 hydrogel NCs exhibited 46% decay in their mass at pH 7.4 (Figure 13B). Meanwhile, at pH 1.2, weight loss was 41% in the case of simple hydrogel formulation (BA6 hydrogels), while, for Ins-Mmt-BA6 hydrogel NCs, only 15% weight loss was seen (Figure 13A).

Overall, less weight loss was observed for hydrogel NC formulations compared to hydrogel formulations at both pH 1.2 and 7.4. This effect is due to restricted swelling of Ins-Mmt-BA6 hydrogel NC preparation compared to BA6 hydrogel because of the presence of crosslinked Ins-Mmt. Moreover, less weight loss was seen for both hydrogel and hydrogel NC formulations at pH 1.2 compared to pH 7.4, where weight loss was much more rapid. This is because at acidic pH, networks remained collapsed and as condensed structrues due to poor penetration of the solvent system within the networks. This poor uptake is associated with a lack of ionization of carboxylic groups into carboxylate ions and, hence, absence of polymeric chain repulsion. Meanwhile, at pH 7.4, carboxylic groups of AA in the form of carboxylate ions cause greater repulsive forces, which results in more swelling of system networks and penetration of surrounding media with resultant rapid degradation. These findings confirm that developed polymeric networks are biodegradable in nature [39,54].

## 4. Conclusions

In the current study, the black seed polysaccharide extract-g-poly (acrylate) pH-responsive hydrogel NCs for safe oral delivery of insulin were successfully formulated by incorporation of insulin-loaded Mmt nanoclay into black seed hydrogels via inward diffusion technique. Black seed hydrogels were prepared earlier by free radical polymerisation from black seed polysaccharide extract, MBA (crosslinker) and AA (monomer for grafting). Black seed hydrogels were also directly loaded with insulin for comparison with hydrogel NCs. FTIR studies confirmed successful grafting of AA on the BSPE backbone and incorporation of insulin in both hydrogel and hydrogel NC formulations without significant denaturation. Thermal studies showed improved thermal stability of hydrogel NCs compared to hydrogel formulations. SEM, EDS and PXRD results also showed the successful incorporation of Ins-Mmt into BA hydrogels to make Ins-Mmt-hydrogel NCs. Swelling, sol-gel fraction and in vitro release results declared the Ins-BA6 hydrogel and Ins-Mmt-BA6 hydrogel NC as the optimum formulations. The acute toxicity studies, carried out for the optimum NC preparation, showed a good safety profile. In vivo studies also presented a decrease in blood glucose levels of diabetic rabbits both by hydrogels and hydrogel NC optimum formulations. Insulin plasma levels of rabbits, measured by the HPLC method, also showed improved pK parameters and prolonged maintenance of blood glucose levels (up to 24 h) for hydrogel NCs compared to respective hydrogel formulations. Therefore, our developed Ins-Mmt-BA6 hydrogel NC formulation (with % RBA of 26.3%) can be a promising tool in the steps moving toward the achievement of safe oral delivery of insulin.

## Figures and Tables

**Figure 1 pharmaceutics-15-00062-f001:**
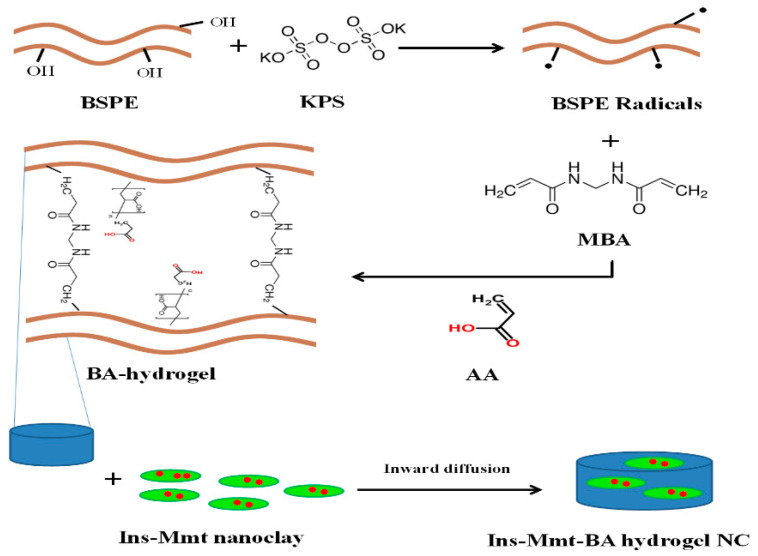
Proposed structure of black seed polysaccharide extract (BSPE)-g-poly (acrylate) hydrogel (BA hydrogel) and insulin intercalated montmorillonite clay (Ins-Mmt)-loaded BA hydrogel nanocomposite (Ins-Mmt-BA hydrogel NC).

**Figure 2 pharmaceutics-15-00062-f002:**
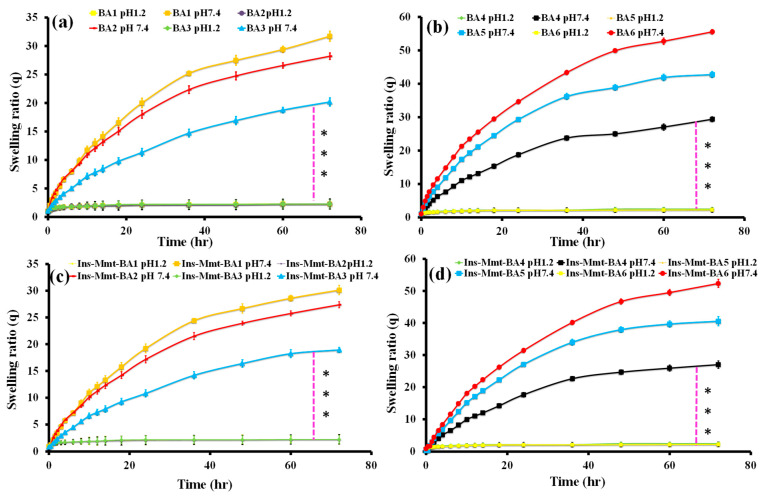
Swelling ratio (*q*) of all hydrogel (BA1-BA6) formulations (**a**,**b**) and all hydrogel NC (Ins-Mmt-BA1 to Ins-Mmt-BA6) formulations (**c**,**d**) at pH 1.2 and pH 7.4. (values are significantly different *** (*p* < 0.001)).

**Figure 3 pharmaceutics-15-00062-f003:**
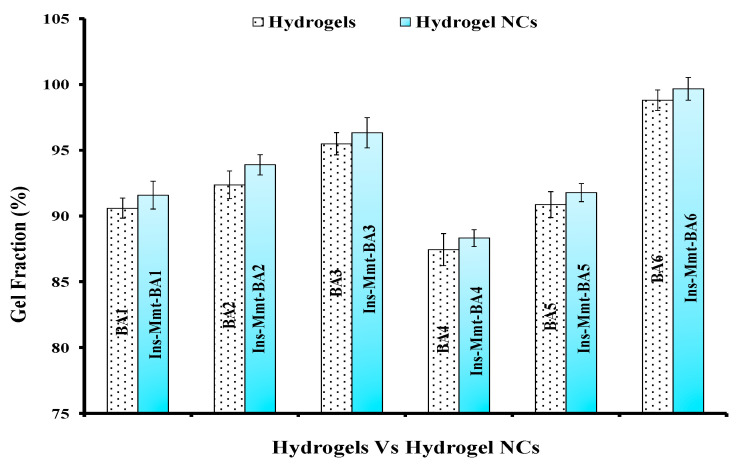
Gel fraction (%) of hydrogels (BA1-BA6) and hydrogel NCs (Ins-Mmt-BA1 to Ins-Mmt-BA6).

**Figure 4 pharmaceutics-15-00062-f004:**
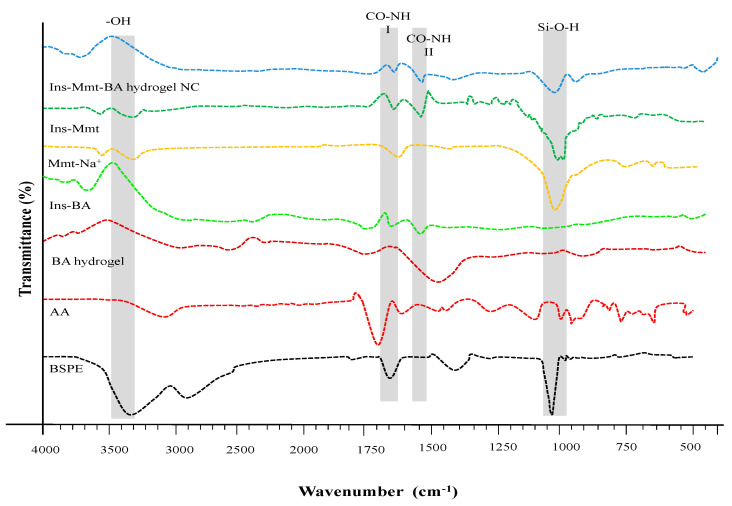
FTIR spectrum overlay of BSPE, AA, BA hydrogel, Mmt – Na^+^ clay, Ins–Mmt, Ins–BA hydrogel and Ins–Mmt–BA hydrogel NC.

**Figure 5 pharmaceutics-15-00062-f005:**
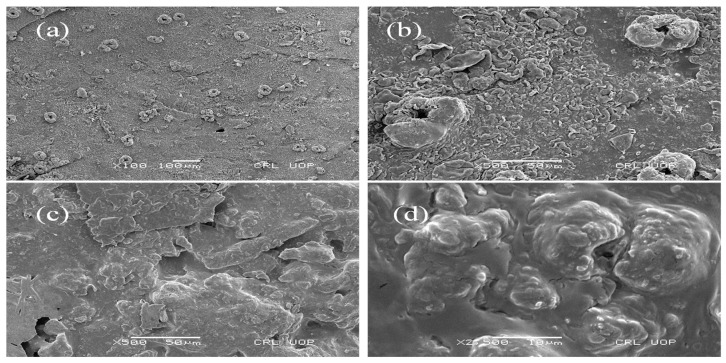
SEM photomicrographs of unloaded BA hydrogel (**a**,**b**) and Ins-Mmt-BA hydrogel NC (**c**,**d**).

**Figure 6 pharmaceutics-15-00062-f006:**
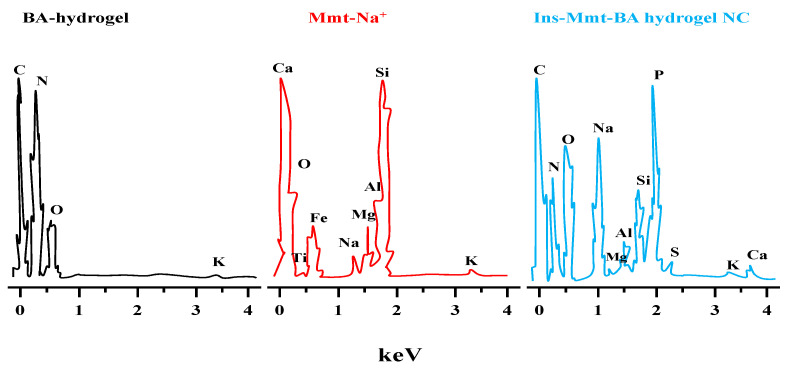
EDS spectra of BA hydrogel, Mmt-Na^+^ clay and Ins-Mmt-BA hydrogel NC.

**Figure 7 pharmaceutics-15-00062-f007:**
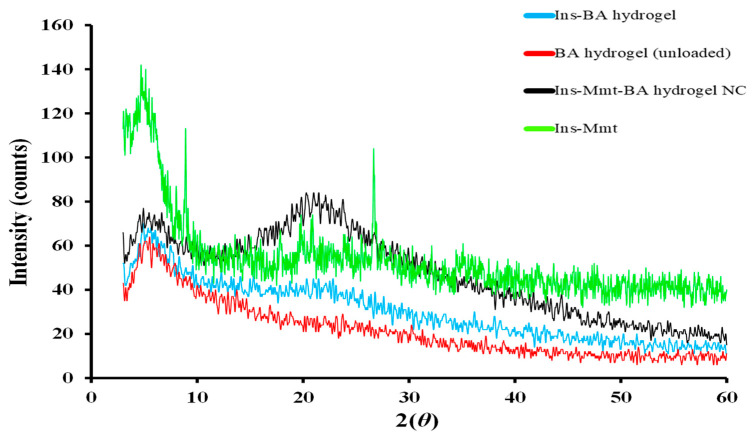
Overlay of X-ray diffractograms of unloaded BA hydrogel, Ins-BA hydrogel, Ins-Mmt-BA hydrogel NCs and Ins-Mmt clay.

**Figure 8 pharmaceutics-15-00062-f008:**
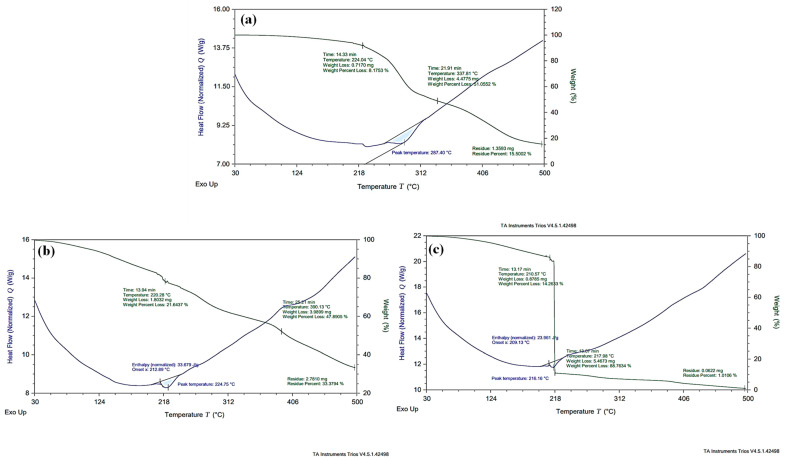
Comparative DLC versus TGA curves of (**a**) unloaded BA hydrogel (**b**) Ins-BA hydrogel (**c**) Ins-Mmt-BA hydrogel NC.

**Figure 9 pharmaceutics-15-00062-f009:**
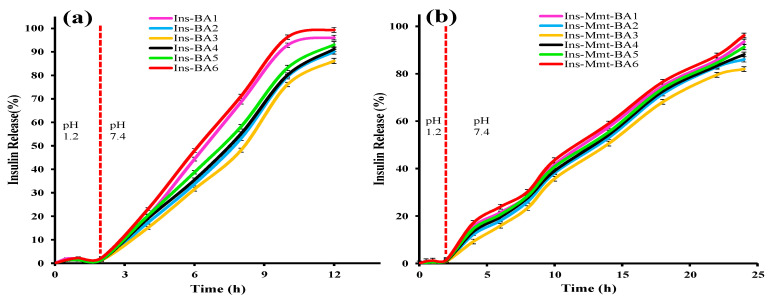
In vitro release profiles of all Ins-BA hydrogel (**a**) and Ins-Mmt-BA hydrogel NC (**b**) formulations with different monomer and crosslinker contents.

**Figure 10 pharmaceutics-15-00062-f010:**
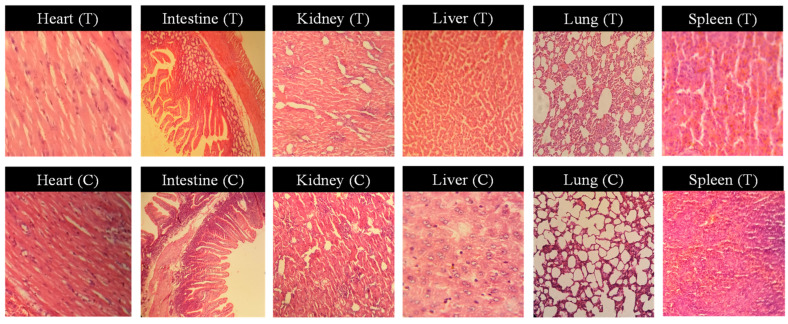
Histological slide representations of vital organs of experimental rabbits, control group (C) and Ins-Mmt-BA hydrogel NC-treated group (T) at 100×. (Analysis date 18 July 2021.)

**Figure 11 pharmaceutics-15-00062-f011:**
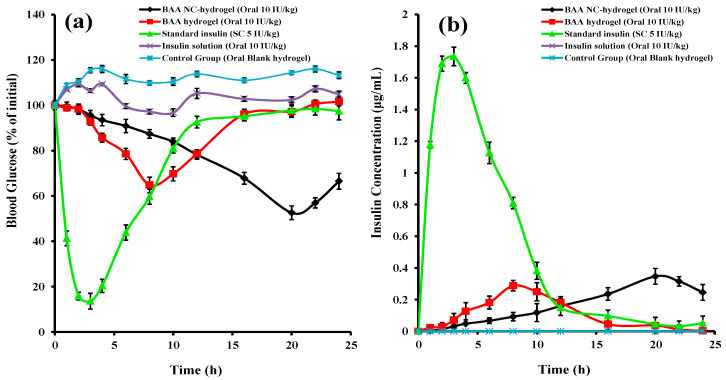
Percentage blood glucose levels of different rabbits’ groups (**a**) and their blood insulin concentrations (**b**) after oral administration of Ins-Mmt-BA hydrogel NC, Ins-BA hydrogel, blank hydrogel (control), insulin solution and SC insulin standard administration.

**Figure 12 pharmaceutics-15-00062-f012:**
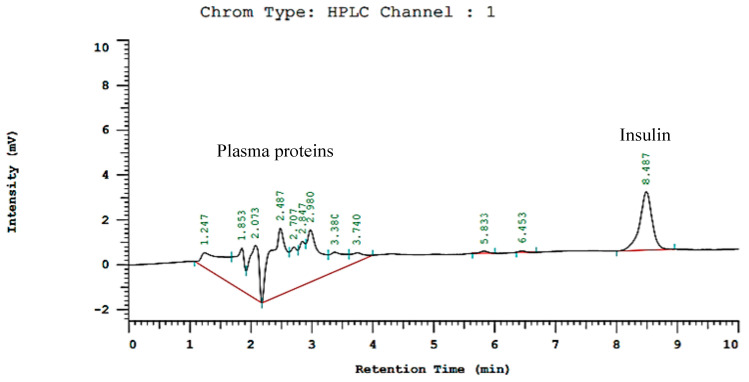
Chromatogram of plasma containing insulin (8.487 min retention time).

**Figure 13 pharmaceutics-15-00062-f013:**
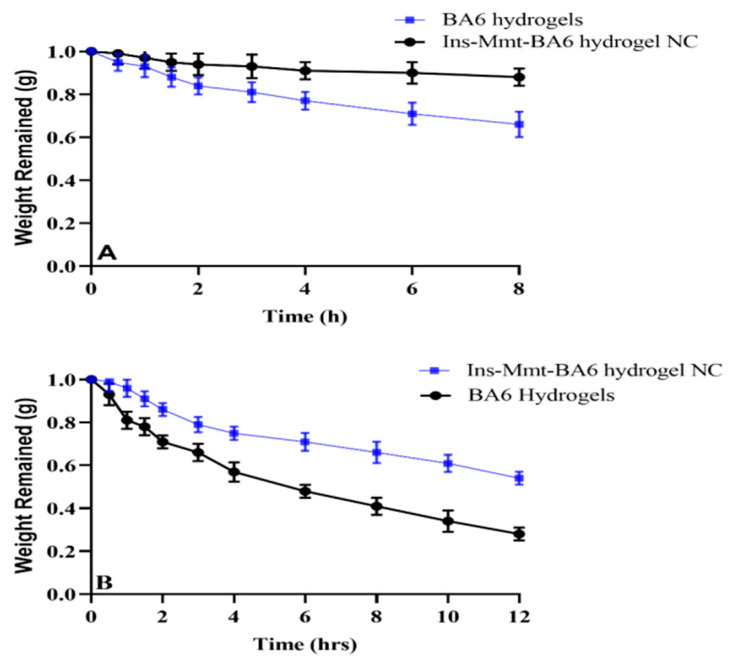
Biodegradation studies at pH 1.2 (**A**) and pH 7.4 (**B**).

**Table 1 pharmaceutics-15-00062-t001:** Composition of black seed polysaccharide (BSPE)-g-poly (acrylate) hydrogel formulations (BA1-BA6) * with varying *N, N*-methylene bisacrylamide (MBA) and acrylic acid (AA) contents.

Formulation Codes	MBA (g/100 g)	AA (g/100 g)
BA1	0.3	15
BA2	0.4	15
BA3	0.5	15
BA4	0.2	10
BA5	0.2	12.5
BA6	0.2	17.5

* Constant quantities of BSPE (1 g/100 g) and potassium persulphate [KPS (0.2 g/100 g)] were used in all the formulations.

**Table 2 pharmaceutics-15-00062-t002:** Results of DLE (%) and maximum swelling ratio (q) for all developed formulations.

Formulation Codes	DLE% ± S.E.M	Maximum Swelling Ratio (q) ± S.E.M
BA1	86.67 ± 1.03	31.72 ± 0.85
BA2	80.53 ± 3.21	28.21 ± 0.73
BA3	70.45 ± 4.42	20.18 ± 0.69
BA4	72.58 ± 2.57	29.38 ± 0.77
BA5	78.91 ± 1.85	42.75 ± 0.81
BA6	90.33 ± 3.15	55.55 ± 0.88
Ins-Mmt-BA1	81.24 ± 2.57	30.12 ± 0.84
Ins-Mmt-BA2	75.47 ± 1.54	27.38 ± 0.79
Ins-Mmt-BA3	66.67 ± 2.33	18.92 ± 0.43
Ins-Mmt-BA4	68.11 ± 3.09	27.02 ± 0.75
Ins-Mmt-BA5	74.37 ± 2.76	40.54 ± 0.82
Ins-Mmt-BA6	85.45 ± 3.51	52.34 ± 0.84

**Table 3 pharmaceutics-15-00062-t003:** Comparative elemental composition (%) of BA hydrogel, Mmt-Na^+^ and Ins-Mmt-BA hydrogel NC.

BA		Mmt-Na^+^		Ins-Mmt-BA	
Elements	Composition (%)	Elements	Composition (%)	Elements	Composition (%)
C	57.32	O	55.81	C	36.27
N	9.14	Mg	0.98	N	8.88
O	33.23	Al	6.18	O	28.94
K	0.31	Na	1.08	Na	7.99
		Si	34.73	Mg	0.27
		K	0.64	Al	1.43
		Ca	0.20	Si	3.99
		Ti	0.25	P	10.31
		Fe	1.22	S	0.35
				K	0.26
				Ca	1.31

**Table 4 pharmaceutics-15-00062-t004:** Physical findings after administration of the developed Ins-Mmt-hydrogel NC (Mean ± S.D, n = 6).

Clinical Findings	Group 1 (Control)	Group 2 (Treated)
Illness Signs	Not Seen	Not Seen
Body weight (kg)		
Pre-treatment	1.76 ± 0.03	1.81 ± 0.02
Day 1	1.83 ± 0.01	1.84 ± 0.03
Day 7	1.81 ± 0.02	1.93 ± 0.01
Day 14	1.91 ± 0.03	1.89 ± 0.03
Food intake (g)		
Pre-treatment	72.33 ± 1.03	74.76 ± 0.99
Day 1	71.56 ± 1.34	73.33 ± 1.23
Day 7	75.32 ± 1.05	72.84 ± 0.98
Day 14	77.29 ± 1.47	75.91 ± 1.17
Water intake (mL)		
Pre-treatment	160.73 ± 1.76	172.88 ± 1.45
Day 1	165.32 ± 1.54	169.71 ± 1.24
Day 7	171.32 ± 1.05	175.32 ± 1.37
Day 14	177.29 ± 1.47	172.92 ± 1.31
Dermal irritation	Not observed	Not observed
Ocular irritation	Not seen	Not seen
Mortality	No	No

**Table 5 pharmaceutics-15-00062-t005:** Haematological findings of rabbit’s blood after Ins-Mmt-hydrogel NC administration (mean ± S.D, n = 6, analysis date 27 May 2021).

Blood Parameter (Unit)	Group 1 (Control)	Group 2 (Treated)
Haemoglobin (g/dL)	11.7 ± 0.47	12.53 ± 0.52
Total RBCs (×10^12^/L)	4.76 ± 0.13	5.11 ± 0.16
Haematocrit (%)	43 ± 0.98	49 ± 0.76
Mean Corpuscular Volume (fL)	75 ± 1.21	69 ± 1.34
Mean Corpuscular Haemoglobin (pg)	25 ± 0.65	23 ± 0.61
Mean Corpuscular Haemoglobin Conc. (g/dL)	27 ± 0.45	31 ± 0.53
Platelet Count (×10^9^/L)	313 ± 3.55	267 ± 3.08
WBC (TLC) (×10^9^/L)	9.2 ± 0.43	7.7 ± 0.51
Neutrophils (%)	62 ± 2.12	70 ± 2.33
Lymphocytes (%)	35 ± 1.23	41 ± 1.52
Eosinophil (%)	02 ± 0.44	03 ± 0.57
Monocytes (%)	03 ± 0.32	04 ± 0.38

**Table 6 pharmaceutics-15-00062-t006:** Renal, hepatic and lipid profiles of rabbits after Ins-Mmt-hydrogel NC administration (mean ± S.D, n = 6; analysis date 27 May 2021).

Biochemical Parameter (Unit)	Group 1 (Control)	Group 2 (Treated)
Bilirubin Total (mg/dL)	0.5 ± 0.01	0.7 ± 0.02
SGPT/ALT (IU/L)	35 ± 1.23	37 ± 1.54
SGOT/AST (IU/L)	31 ± 1.12	34 ± 1.09
Alkaline Phosphatase (U/L)	103 ± 2.46	111 ± 3.51
Urea (mg/dL)	34 ± 1.29	36 ± 1.48
Creatinine (mg/dL)	0.9 ± 0.05	0.8 ± 0.04
Uric Acid (mg/dL)	5.6 ± 0.27	4.9 ± 0.19
Triglycerides (mg/dL)	78 ± 2.18	72 ± 2.05

**Table 7 pharmaceutics-15-00062-t007:** Pharmacokinetic parameters of orally administered Ins-BA hydrogel, Ins-Mmt-BA hydrogel NC and SC standard insulin (values are significantly different * (*p* < 0.05)).

Formulation *	pK Parameters							
	C_max_ (µg/mL)	t_max_ (h)	t_1/2_ (h)	MRT (h)	(AUC)_0–∞_ (µg h/mL)	Cl (mL/h)	Vd (mL)	RBA (%)
SC insulin standard	1.735	3	3.493	6.007	12.985	13.361	67.326	
Oral Ins-BA hydrogel	0.288	8	1.065	9.608	2.616	132.630	203.83	10.07
Oral Ins-Mmt-BA hydrogel NC	0.347	20	7.978	24.390	6.831	50.799	584.66	26.3

## Data Availability

Research data can be provided by corresponding authors on reasonable request.

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
