# Peer review of "Novel Black Seed Polysaccharide Extract-g-Poly (Acrylate) pH-Responsive Hydrogel Nanocomposites for Safe Oral Insulin Delivery: Development, In Vitro, In Vivo and Toxicological Evaluation"

_pharmaceutics, 2022, doi:10.3390/pharmaceutics15010062_

Round 1

Reviewer 1 Report

Authors proposed a paper entitled “Novel black seed polysaccharide extract-g-poly (acrylate) pH-responsive hydrogel nanocomposites for safe oral insulin delivery: Development, in-vitro, in-vivo and toxicological evaluation” for the publication in Pharmaceutics, mdpi.

This paper is characterized by a good use of English and a quite interesting scientific soundness.

Authors should insert additional references to their paper, in order to support their explanations.

I only asked some revisions in this paper, as follows:

I suggest adding an abbreviation list, according to the guidelines of this journal.

“DM1 is a chronic autoimmune disease” information correctly reported need a reference as support.

“The clinical onset of disease reflect” remove double space before this sentence.

“So, there is a dir” please, use more formal expressions as “therefore” instead of “so”.

“can destroy many proteinaceous drugs like insulin.” Please add reference as support.

nanomaterials are widely employed in pharmaceutical sector due to their nano-dimensions (less” I think it is quite pleonastic and obvious saying that nanomaterials are employed due to nano dimensions. There are several advantages deriving by nano- morphology. Please city those advantages with proper references as support.

Pag. 4. “After 24 h,” remove double space before it.

Table 1 acronyms should be defined

Section 2.9. Information about gold covering of particles and sputter coater model should be provided.

I suggest modify the proportions of figure 1. Please reduce the dimensions along x-axis .

“(88.31%-99.67%)It was again due to the same reason of increased number of crosslink point with more MBA”. Please add a full stop after the parenthesis and before “it was due to… Moreover, I suggest adding references to this explained reason.

Please, improve Table 3 according to the guidelines of this Journal.

Section 3.14, “The reason is the destruction of insulin at gastric environment due to gastric enzymes” please add reference here.

Author Response

REVIEWER 1

Authors proposed a paper entitled “Novel black seed polysaccharide extract-g-poly (acrylate) pH-responsive hydrogel nanocomposites for safe oral insulin delivery: Development, in-vitro, in-vivo and toxicological evaluation” for the publication in Pharmaceutics, MDPI.

This paper is characterized by a good use of English and a quite interesting scientific soundness.

Authors should insert additional references to their paper, in order to support their explanations.

I only asked some revisions in this paper, as follows:

  1. I suggest adding an abbreviation list, according to the guidelines of this journal.

Response: Thank you very much for your suggestion. We made changes to mention each abbreviation in parenthesis upon its first use in abstract, main text and captions of figures and tables.

  1. “DM1 is a chronic autoimmune disease” information correctly reported need a reference as support.

Response: Thank for your comment. Required reference has now been added in revised version of manuscript. Following reference has been added to support that DM1 is a chronic autoimmune disease;

Kalousová M, Fialová L, Skrha J, Zima T, Soukupová J, Malbohan IM, Stipek S. Oxidative stress, inflammation and autoimmune reaction in type 1 and type 2 diabetes mellitus. Prague Med Rep. 2004 Jan 1;105(1):21-8.

  1. “The clinical onset of disease reflect” remove double space before this sentence.

Response: The double space has now been removed in the section 1 (Introduction, paragraph 2, line 8).

  1. “So, there is a dir” please, use more formal expressions as “therefore” instead of “so”.

Response: Thank you again for pointing out this mistake, The sentence in section 1 (Introduction, paragraph 2, line 10) has been rephrased as per our kind suggestion in revised version of manuscript.

  1. “can destroy many proteinaceous drugs like insulin.” Please add reference as support.

Response: Thank you for your query. Following reference at number 6 has been added to support

                 this sentence in revised version of manuscript;

Gedawy A, Martinez J, Al-Salami H, Dass CR. Oral insulin delivery: existing barriers and current counter-strategies. Journal of pharmacy and pharmacology. 2018 Feb;70(2):197-213.

  1. “nanomaterials are widely employed in pharmaceutical sector due to their nano-dimensions (less” I think it is quite pleonastic and obvious saying that nanomaterials are employed due to nano dimensions. There are several advantages deriving by nano- morphology. Please city those advantages with proper references as support.

Response: Thank you very much for this correction. We have changed this sentence.  Moreover, we have also added an article containing the nanomaterial derived advantages with reference number at 15 number reference within the introduction section. Here is the reference;

Baig N, Kammakakam I, Falath W. Nanomaterials: A review of synthesis methods, properties, recent progress, and challenges. Materials Advances. 2021;2(6):1821-71.

  1. 4. “After 24 h,” remove double space before it.

Response: The double space has now been removed.

  1. Table 1 acronyms should be defined

Response: Table 1 acronyms has now been defined as per your kind suggestion in revised version of manuscript.

  1. Section 2.9. Information about gold covering of particles and sputter coater model should be provided.

Response: Thank you for your query. Require information has now been added in the section 2.9 of the revised manuscript.

  1. I suggest modify the proportions of figure 1. Please reduce the dimensions along x-axis.

Response: Thank for your kind suggestion in terms of modification of figure 1 proportions. We have reduced the x-axis dimensions. Now it’s more presentable. Thank you.

  1. “(88.31%-99.67%)It was again due to the same reason of increased number of crosslink point with more MBA”. Please add a full stop after the parenthesis and before “it was due to… Moreover, I suggest adding references to this explained reason.

Response: Thank you for reviewing manuscript critically and closely. Changes are done as you have pointed out. Moreover, relevant references have now been added at reference no. 30 and 39 within the revised manuscript.

  1. Please, improve Table 3 according to the guidelines of this Journal.

Response: Thank you for your suggestion. Table 3 has now been improved according to the Pharmaceutics journal guidelines in revised version of manuscript.

  1. Section 3.14, “The reason is the destruction of insulin at gastric environment due to gastric enzymes” please add reference here.

Response: Thank you for your query. The reference has now been added in the main text at the required place. Reference number 6 is the supporting reference.

Reviewer 2 Report

This study is an interesting approach to oral delivery of peptides and specifically insulin. The use of natural polysacharides to this end is trending lately and though a lot of work is still required to reach clinical applications, such efforts are promising. My points are given below:

-Some abbreviations are missing in the abstract.
-I do not condone the use of abbreviations in keywords.
-Figures and Tables are too many so maybe some can be moved to supplementary materials, eg Table 4.
-Figures 2 and 8 quality and readability are not good. Tags inside should be of bigger font size.

INTRO
-The sentence "Improved...form" is not clear. What does PK profile have to do with compliance and trust?

METHODS
-Did the authors consider to evaluate particle size, homogeneity and charge through light scattering techniques?
-Utilized software and details are missing in some methods, particularly 2.8-2.12.

RESULTS
-3.3: What is the relevance of hydrogel sweling with biological applications? How are the properties going to affect in vivo application? In addition, would the hydrogels continue to swallo over time? A plateau does not seem to have been achieved
-3.4: Isnt 214nm too low to measure insulin? Could the authors have gone higher to avoid background?
-3.9: How do the authors support the crystallinity of insulin in the various cases?
-3.10: For DSC, if the 225 peak is attributed to insulin, then where did the 287.4 peak of the hydrogel go? Then again, if the peak at 216 is hydrogel decomposition, where did the insulin peak go? Obvisously, these singular peaks are of the whole system in each case. In addition, do we expect insulin denaturation at some point?
-3.11: The authors should correlate the observed release behavior with the expected biological behavior.
-3.13: The analysis day should be mentioned in all table and figure legends, ie Tables 6,7 and Figure 7. Also, from the images in Figure 7, there seems to be an altered moprhology of tissue in all cases after treatment, especially in liver, lung and spleen. Are these findings important?
-3.14: Why was the 5th group administered with only 5IUs?
-3.16: I believe that the in vitro study should be moved before the oral toxicity study, smae with the method.

CONCLUSIONS
-Though the rest of the results are promising, PK parameters were not overall improved for the developed formulations compared to standard treatment. Indeed there was a prolonged effect, but parameters like AUC and Cmax were much lower than the standard treatment. This study is a step forward towards oral peptide delivery but there is still a long way to go.

-Though permissions for in vivo studies are mentioned inside the manuscript, I believe that there should be also a separate statement, after the main text.

Author Response

Reviewer 2

Comments and Suggestions for Authors

This study is an interesting approach to oral delivery of peptides and specifically insulin. The use of natural polysacharides to this end is trending lately and though a lot of work is still required to reach clinical applications, such efforts are promising. My points are given below:

  1. -Some abbreviations are missing in the abstract.

Response: Thank you very much for pointing out this point. In revised version, we have incorporated missing abbreviations as per your kind suggestion.

  1. -I do not condone the use of abbreviations in keywords.

Response: Thank you sir for the suggestion. We have removed the abbreviations from the list of keywords in revised version of manuscript.

  1. -Figures and Tables are too many so maybe some can be moved to supplementary materials, eg Table 4.

Response: Thank you for your suggestion. We have added possible Table or Figure into supplementary data section like Table 4.

  1. -Figures 2 and 8 quality and readability are not good. Tags inside should be of bigger font size.

Response: Thank you for your comment. In order to improve quality and readability font size of tags in Figure 2 and Figure 8 has now been increased.

INTRO

  1. -The sentence "Improved...form" is not clear. What does PK profile have to do with compliance and trust?

Response: Thank you for your comment. The subcutaneous regular insulin which requires multiple administration in 24 hrs (after every 8-12 hrs)*. If a novel oral dosage form with improved pK-profile can provide sustained release over a period of 24 hrs with once daily dose administration. Then definitely patient compliance and trust on that novel dosage form will be increased as compared to the dosage form requiring 2-3 times daily administration via invasive parenteral route.

*(https://www.icliniq.com/articles/drug-and-supplements/insulin-timing#:~:text=A%20rapid-acting%20insulin%20is%20administered%20every%203%20to,It%20Be%20Fine%20to%20Take%20Insulin%20After%20Food%3F).

METHODS

  1. -Did the authors consider to evaluate particle size, homogeneity and charge through light scattering techniques?

Response: The dynamic light scattering technique is a good option for particle size determination. It requires liquid suspension samples. (https://doi.org/10.1016/B978-0-12-814182-3.00010-9). But in case of our developed hydrogel nanocomposite discs, the insulin intercalated montmorillonite-Na+ nanoclay (Ins-Mmt) is crosslinked in the formulation. Thus it becomes a part of formulation providing mechanical strength to hydrogel nanocomposite. It doesn’t release itself in dissolution medium but only releases insulin. Therefore required suspension sample can’t be made. Moreover, this system must not be confused with polymeric micro or nanoparticles those require such type of characterization.

We used energy dispersive X-Ray (EDX) technique instead to determine and compare elemental composition of unloaded hydrogel, and Ins-Mmt incorporated hydrogel nanocomposite formulations.

  1. -Utilized software and details are missing in some methods, particularly 2.8-2.12.

Response: Thank you for your kind suggestion. Software details have now been mentioned in the respective sections as per your suggestion.

RESULTS

  1. -3.3: What is the relevance of hydrogel swelling with biological applications? How are the properties going to affect in vivo application? In addition, would the hydrogels continue to swell over time? A plateau does not seem to have been achieved

Response: This is a good question. In the section 3.4 we have mentioned the direct relationship of hydrogel swelling with percentage drug loading efficiency (%DLE). The more the hydrogel was swollen, more the % DLE was observed. Subsequently, the same direct relationship was observed with the % DLE and % release of the insulin in developed hydrogel and hydrogel nanocomposite (NC) formulations. It means; >hydrogel swelling > % DLE > % insulin release in-vitro. Later on, we selected the best % insulin release formulations for in-vivo studies from both hydrogel and hydrogel NC formulations.

Moreover, the developed hydrogel and hydrogel NC formulations, both showed pH- responsive behavior in-vitro. They showed less than 3% swelling in 2 hrs at pH 1.2. The same formulation discs showed much higher rate of swelling and drug release (Fig 2 & Fig 9) when subjected to the pH 7.4. This pH-responsive swelling and release behavior has a direct impact on in-vivo insulin release and bioavailability by preventing insulin release at gastric pH and allowing its release in intestinal environment.

The second part of the question is about hydrogel swelling over time. We conducted the swelling studies till 72 hrs. Though the hydrogels kept on swelling but the rate of swelling was much lower as compared to first 24 h (Fig 2)

  1. -3.4: Isnt 214nm too low to measure insulin? Could the authors have gone higher to avoid background?

Response: thank you for your valuable query. In many previously reported studies, the UV-visible wavelength for insulin detection is mentioned as 214 nm.

(https://doi.org/10.1016/j.jpba.2005.10.016 and https://doi.org/10.1016/j.ijbiomac.2020.01.302)

  1. -3.9: How do the authors support the crystallinity of insulin in the various cases?

Response: Sir in section 3.9, we have added the reference of Bueche, B. (2013) thesis (http://hdl.handle.net/10454/5688). Here, on page 14-15 of his thesis, he shared the XRD pattern along with SEM images of pure insulin crystals with 5% moisture. The XRD pattern of pure insulin crystals, (specially a broad peak around 15° to 25° angles) quite resembles as our PXRD pattern findings. The author compared the results of SEM (clear cut crystal structure observed) and XRD pattern (no sharp crystalline peaks observed), as also observed in our study results (Fig. 7). The Bueche further clarifies that the reason is the % moisture less than 10 % in the XRD samples that causes X-ray artifacts as protein crystallographers usually mount crystals from saturated solutions (~30% RH) for analysis.

  1. -3.10: For DSC, if the 225 peak is attributed to insulin, then where did the 287.4 peak of the hydrogel go? Then again, if the peak at 216 is hydrogel decomposition, where did the insulin peak go? Obvisously, these singular peaks are of the whole system in each case. In addition, do we expect insulin denaturation at some point?

Response: Thank you very much for your critical analysis. Yes Sir you are right that these singular peaks are attributed to the whole system and specifically with these thermal results, we can only tell that which insulin-loaded system (either hydrogel or hydrogel NC) showed overall better thermal stability from room temperature to 200 °C. But the solely insulin denaturation/stability can’t be detected with these results. In previous literature, Li et al. (2019) has worked on metamaterial terahertz sensor for measuring thermal-induced denaturation temperature of insulin (https://doi.org/10.1109/JSEN.2019.2949617). According to their findings, the denaturation of insulin starts from 70 °C and completes at about 88°C, and these two temperatures are independent of the initial concentrations of the insulin.

We gratefully acknowledge your opinion. We are modifying the discussion of section 3.10 by excluding “insulin stability” statement and focusing only on the comparison of thermal stability of insulin loaded BA-hydrogel formulation and respective Ins-Mmt hydrogel NC formulation.

  1. -3.11: The authors should correlate the observed release behavior with the expected biological behavior.

Response: Thank you Sir for the valuable suggestion. We have included the correlation between observed release behavior and its possible impact on biological behavior at the end of section 3.11 under the reference 6 in the main text of revised version of manuscript.

  1. -3.13: The analysis day should be mentioned in all table and figure legends, ie Tables 6,7 and Figure 7. Also, from the images in Figure 7, there seems to be an altered moprhology of tissue in all cases after treatment, especially in liver, lung and spleen. Are these findings important?

Response: Analysis date has been mentioned in the respective figure and tables legends as per advice. After in-vivo pK and bioavailability studies, the alloxan-treated rabbits were further investigated for acute toxicity. It is common effect of alloxan to induce morphological changes to liver, kidney and spleen.

  1. -3.14: Why was the 5th group administered with only 5IUs?

Response: The 5th group of rabbits was administered with subcutaneous standard dose of insulin. If it was administered with larger doses, then marked lethal hypoglycemia could be observed in experimental animals.

  1. -3.16: I believe that the in vitro study should be moved before the oral toxicity study, same with the method.

Response: Thank you for your kind suggestion we have moved the in-vitro studies before the oral toxicity studies in the main text within revised version of manuscript.

CONCLUSIONS

  1. -Though the rest of the results are promising, PK parameters were not overall improved for the developed formulations compared to standard treatment. Indeed there was a prolonged effect, but parameters like AUC and Cmax were much lower than the standard treatment. This study is a step forward towards oral peptide delivery but there is still a long way to go.

Response: Thank you for your valuable comments and appreciation. Sir we have put our efforts in order to optimized both type of hydrogels by tuning their contents. Confirmed their compatibility with living systems. Carried out in-vivo studies just to explore pK parameters. Although results in this regard are not promising in terms of Cmax and AUC but prolonged effect is there. Moreover, in future we can re-validate these finding by modifying chromatographic conditions.

  1. -Though permissions for in vivo studies are mentioned inside the manuscript, I believe that there should be also a separate statement, after the main text.

Response: Thank you for your kind suggestion. We have included the in-vivo studies permissions as a separate statement after the main text.

Reviewer 3 Report

The paper entitled ”Novel black seed polysaccharide extract-g-poly (acrylate) pH-responsive hydrogel nanocomposites for safe oral insulin delivery: Development, in-vitro, in-vivo and toxicological evaluation” by Farya Shabir et al. describes the application of Nanogel-based drug carriers for the treatment of diabetes through oral delivery. This work demonstrates the big potential of nanodrug delivery systems to improve the way we practice medicine.  Overall, this is a clear, concise, and well-written manuscript. Specific comments follow.

Introduction:

1.       Sufficient information about the previous study findings is required for readers to follow the present study rationale and procedures.

2.       The author state that “Currently nanomaterials are widely employed in pharmaceutical sector due to their nano-dimensions (less than 100 nm)” there are plenty of FDA-approved nanocarriers which are bigger than 100 nm.  Therefore, the success of nanoparticles in the pharmaceutical sector is not because their size is less than 100 nm.

EXPERIMENTAL – Materials

3.       It was mentioned that Nigella sativa seeds (black seed/kalonji) were purchased from a regional market with no Cat No / company name. Since it wasn’t produced by a scientific supplier, how can we rely on this uncharacterized material?

Results

4.       Figures 2 and 8 are blurred with unsatisfied resolution.

5.       I couldn’t see anywhere the authors point out the size range of prepared nanogels.

Author Response

Comments and Suggestions for Authors

The paper entitled ”Novel black seed polysaccharide extract-g-poly (acrylate) pH-responsive hydrogel nanocomposites for safe oral insulin delivery: Development, in-vitro, in-vivo and toxicological evaluation” by Farya Shabir et al. describes the application of Nanogel-based drug carriers for the treatment of diabetes through oral delivery. This work demonstrates the big potential of nanodrug delivery systems to improve the way we practice medicine.  Overall, this is a clear, concise, and well-written manuscript. Specific comments follow.

Introduction:

  1. Sufficient information about the previous study findings is required for readers to follow the present study rationale and procedures.

Response: Thank you very much for highlighting this deficiency. We have incorporated a paragraph related to previous study findings in the revised manuscript under reference number 9. 

  1. The author state that “Currently nanomaterials are widely employed in pharmaceutical sector due to their nano-dimensions (less than 100 nm)” there are plenty of FDA-approved nanocarriers which are bigger than 100 nm.  Therefore, the success of nanoparticles in the pharmaceutical sector is not because their size is less than 100 nm.

Response: Thank you for your valuable suggestion. In order to comply your valuable comment the size specifications have now been removed in revised version of manuscript.

EXPERIMENTAL – Materials

  1. It was mentioned that Nigella sativa seeds (black seed/kalonji) were purchased from a regional market with no Cat No / company name. Since it wasn’t produced by a scientific supplier, how can we rely on this uncharacterized material?

Response: All the seeds were physically and carefully inspected for size, shape and for presence of any other type of seeds. In order to further confirm their quality they were subjected to water test. In this test seeds were soaked in a wide beaker containing water and observation was made for 1 to 2 hrs. All the seeds were remained in sink state and no one was floating thus ensuring that seeds were of good quality.

  1. Figures 2 and 8 are blurred with unsatisfied resolution.

Response: As per your kind suggestion, the figures with better resolution have now been added in revised version of the manuscript.

  1. I couldn’t see anywhere the authors point out the size range of prepared nanogels.

Response: We thank the reviewer from the bottom of our hearts for these very constructive criticisms that we took into consideration to improve the paper and make it more accessible to readers. That said, we have clarified some points that we naively believe must have been misinterpreted or misexplained. We regret not being able to explain that this work is not a drug delivery system based on nanomaterials and that it was not submitted as a work that aims to use nanomaterials. We had to explain to our dear readers that it was entirely a system based on polymers interconnected via a crosslinking agent. Such a microstructure retains its geometry even after imbibition of the biological fluid.

Round 2

Reviewer 1 Report

Authors provided a revised version of their paper.

Authors responded point by point to all issues raised by reviewers.

The quality of the paper improved.

I only suggest to check the new sentence reported as “In earlier studies the many biodegradable polymeric systems both natural and syn-thetic have been tried to enhance oral bioavailability of insulin”. I would change it with “In previous studies, the use of several natural and synthetic biodegradable polymeric systems has been attempted to enhance…”

“subsequently placed on the aluminum stud” please replace “stud” with “stub”.

The focus of figures improved significantly.

In the sentence ending with “…in-vivo protection of insulin from enzymatic degradation and bioavailability” in-vivo should be in italique.

Author Response

Comments and Suggestions for Authors

Authors provided a revised version of their paper.

Authors responded point by point to all issues raised by reviewers.

The quality of the paper improved.

I only suggest to check the new sentence reported as “In earlier studies the many biodegradable polymeric systems both natural and syn-thetic have been tried to enhance oral bioavailability of insulin”. I would change it with “In previous studies, the use of several natural and synthetic biodegradable polymeric systems has been attempted to enhance…”

Response: We appreciate remarkable review reports of the reviewers in Round One and also in round two. Thank You for your change. It has now been made in revised version of manuscript.  

“subsequently placed on the aluminum stud” please replace “stud” with “stub”.

Response:  Thank you for this typo error. Correction has now been made in revised version of manuscript. 

The focus of figures improved significantly.

Response: Thank you for your valuable comment. 

In the sentence ending with “…in-vivo protection of insulin from enzymatic degradation and bioavailability” in-vivo should be in italique.

Response:  Once again thank you for highlighting this point. We have done this formatting change in the revised version of manuscript.